# Crystal structure and catalytic mechanism of PL35 family glycosaminoglycan lyases with an ultrabroad substrate spectrum

Lin Wei[1†], Hai-Yan Cao[2†], Ruyi Zou[1], Min Du[1], Qingdong Zhang[1,3], Danrong Lu[1,3], Xiangyu Xu[1], Yingying Xu[1], Wenshuang Wang[1], Xiu-Lan Chen[4,5], Yu-Zhong Zhang[2,4,5]*, Fuchuan Li[1,5]*

[1]National Glycoengineering Research Center and Shandong Key Laboratory of Carbohydrate Chemistry and Glycobiology, State Key Laboratory of Microbial Technology, Shandong University, Qingdao, China; [2]MOE Key Laboratory of Evolution and Marine Biodiversity, Frontiers Science Center for Deep Ocean Multispheres and Earth System & College of Marine Life Sciences, Ocean University of China, Qingdao, China; [3]School of Life Science and Technology, Weifang Medical University, Weifang, China; [4]Marine Biotechnology Research Center, State Key Laboratory of Microbial Technology, Shandong University, Qingdao, China; [5]Joint Research Center for Marine Microbial Science and Technology, Shandong University and Ocean University of China, Qingdao, China

*For correspondence:
zhangyz@sdu.edu.cn (Y-ZZ);
fuchuanli@sdu.edu.cn (FL)

†These authors contributed equally to this work

Competing interest: The authors declare that no competing interests exist.

## eLife Assessment

This **useful** manuscript reports on the crystal structures of two glycosaminoglycan (GAG) lyases from the PL35 family, along with in vitro enzyme activity assays and comprehensive structure-guided mutagenesis. The authors have addressed key concerns by incorporating additional docking analyses, validating the role of His188 in alginate degradation, and providing ICP-MS data to examine $Mn^{2+}$ binding. While these improvements enhance the study, the study is **incomplete** due to the lack of enzyme-substrate complex structures and reliance on modeling which still limit mechanistic insight. Nonetheless, the revised manuscript presents a more complete analysis that will be of interest to specialists in carbohydrate-active enzymes.

**Abstract** Recently, a new class of glycosaminoglycan (GAG) lyases (GAGases) belonging to PL35 family has been discovered with an ultrabroad substrate spectrum that can degrade three types of uronic acid-containing GAGs (hyaluronic acid, chondroitin sulfate and heparan sulfate) or even alginate. In this study, the structures of GAGase II from *Spirosoma fluviale* and GAGase VII from *Bacteroides intestinalis* DSM 17393 were determined at 1.9 and 2.4 Å resolution, respectively, and their catalytic mechanism was investigated by the site-directed mutant of their crucial residues and molecular docking assay. Structural analysis showed that GAGase II and GAGase VII consist of an N-terminal $(\alpha/\alpha)_6$ toroid multidomain and a C-terminal two-layered β-sheet domain with $Mn^{2+}$. Notably, although GAGases share similar folds and catalytic mechanisms with some GAG lyases and alginate lyases, they exhibit higher structural similarity with alginate lyases than GAG lyases, which may present a crucial structural evidence for the speculation that GAG lyases with $(\alpha/\alpha)_n$ toroid and antiparallel β-sheet structures arrived by a divergent evolution from alginate lyases with the same folds. Overall, this study not only solved the structure of PL35 GAG lyases for the first time and investigated their catalytic mechanism, especially the reason why GAGase III can additionally degrade alginate, but also provided a key clue in the divergent evolution of GAG lyases that originated from alginate lyases.

## Introduction

GAGs are a class of linear polyanionic polysaccharides ubiquitously distributed on the cell surface and in the extracellular matrix (ECM) of animal tissues (*Cohen and Merzendorfer, 2019*), and they participate in various physiological and pathological processes through interacting with a series of chemokines (*Dyer et al., 2016*; *Irie et al., 2008*), growth factors (*Sirko et al., 2010*; *Zhang et al., 2019*) or other ECM components. Except for keratan sulfate (KS), which does not contain hexuronic acid (HexUA) residues, other GAGs are composed of repeating disaccharides consisting of HexUA (D-glucuronic acid (GlcUA) or L-iduronic acid (IdoUA)) and hexosamine (HexNAc) (D-N-acetylgalactosamine (GalNAc) or D-N-acetylglucosamine (GlcNAc)). Based on the disaccharide composition and the type of glycosidic bonds between disaccharides units, HexUA-containing GAGs are classified into three classes: hyaluronan (HA), chondroitin sulfate (CS)/dermatan sulfate (DS) and heparan sulfate (HS)/heparin (Hep) (*Cohen and Merzendorfer, 2019*). HA is the only non-sulfated GAG composed of repeating -GlcUAβ1-3GlcNAc- disaccharides linked by β1–4 glycosidic bonds. In contrast, the structures of CS/DS and Hep/HS are highly complex due to sulfation and epimerization modifications. The backbone of CS is composed of repeating -GlcUAβ1-3GalNAc- disaccharides linked by β1–4 glycosidic bonds, and is further sulfated at C-2 of GlcUA and C-4/C-6 of GalNAc by sulfotransferases; meanwhile, the GlcUA residues in CS can be epimerized into IdoUA residues by glucuronyl C-5 epimerase to form DS (*Kusche-Gullberg and Kjellén, 2003*). Similarly, HS/Hep composed of repeating –4GlcUAβ1-4GlcNAcα1- units can be sulfated at C-2 of GlcUA and N/C-2/C-3/C-6 of GlcNAc/deacetylated GlcN, and GlcUA residues are often epimerized to IdoUA residues, especially in Hep (*Cohen and Merzendorfer, 2019*). Such complex modification through sulfation and epimerization endows sulfated GAGs with diverse biological functions.

GAG lyases, belonging to the superfamily of polysaccharide lyases (PLs), specifically catalyze the degradation of HexUA-containing GAGs (*Charnock et al., 2002*; *Garron and Cygler, 2010*). These reactions generate a plethora of oligosaccharides that contain an unsaturated double bond between C-4 and C-5 on uronic acids at their nonreducing ends, which is generated via a β-elimination mechanism (*Garron and Cygler, 2010*). As a complimentary mechanistic strategy to glycoside hydrolases (GHs), GAG lyases degrade HexUA-containing GAGs without the participation of a water molecule and are involved in the polysaccharide metabolism of microorganisms (*Garron and Cygler, 2010*; *Ndeh et al., 2020*). For the past many years, an increasing number of gene sequences containing PL catalytic modules and some ancillary modules have been annotated as PLs (*Lombard et al., 2010*). Based on their sequence similarity, 44 families (and an unclassified PL0 family) have been hierarchically classified in the CAZy database (http://www.cazy.org/), and at least one member of each family has been subjected to activity identification and biochemical property analysis (*Drula et al., 2022*). The identified GAG lyases are widely distributed in the PL6, PL8, PL12, PL13, PL15, PL16, PL21, PL23, PL29, PL30, PL33, PL35, and PL37 families (*Drula et al., 2022*). Based on substrate specificity, GAG lyases are classified as HA-specific lyases, which can degrade HA only, CS/DS lyases (chondroitinase, CSases), which degrade CS/DS as well as HA in most cases, and Hep/HS lyases (heparinases, Hepases), which specifically cleave Hep/HS. All identified GAG lyases have substrate specificity to strictly distinguish one or two kinds of GAGs based on saccharide composition and glycosidic bonds between each HexUA-HexNAc disaccharide units, while other uronic acid-containing polysaccharides are usually unable to serve as substrates for these enzymes.

With the evolution of host polysaccharides (*Csoka and Stern, 2013*; *Popper et al., 2011*), the bacterial enzymes that degrade these polysaccharides have also evolved to varying degrees in response to complex environments. As a kind of unbranched anionic extracellular polysaccharide produced by lower organisms algae and bacteria such as brown alga (*Popper et al., 2011*) and bacteria belonging to *Pseudomonas* (*Evans and Linker, 1973*) and *Azotobacter* (*Clementi, 1997*), alginate containing no hexosamine and sulfation possesses a simpler structure than GAGs in animals and likely emerge earlier than GAGs during the course of evolution. Like GAGs, alginate is a linear polyanionic polysaccharide that contains HexUA residues; however, alginate possesses a completely different chemical structure, which is composed of D-mannuronate (M) and its C5 epimer L-guluronate (G), and M and G residues often alternate randomly to form heteropolyuronic blocks, including polyM composed of M residues linked by β1–4 bonds, polyG composed of G residues linked by α1–4 bonds, and polyMG/GM composed of alternating M and G linked by β/α1–4 bonds (*Pawar and Edgar, 2012*). Likewise, a series of alginate lyases essential for the metabolism of alginate were identified

from microorganisms and algae as well as lower marine animals, which belong to the PL5, PL6, PL7, PL8, PL14, PL15, PL17, PL18, PL31, PL32, PL34, PL36, PL38, PL39, PL41, and PL44 families (*Drula et al., 2022*). Based on their specificity, these lyases are also divided into the following categories: M block-specific lyases (*Itoh et al., 2019*; *Zhu et al., 2015*), G block-specific lyases (*Matsubara et al., 1998*; *Yamasaki et al., 2005*) and MG-specific alginate lyases (*Jagtap et al., 2014*; *Yamasaki et al., 2004*). From an evolutionary perspective in previous reports, GAG lyases are thought to have originated from alginate lyases by a divergent evolution through adaptation to the evolution of substrate polysaccharides. Some similarities in the sequences and folds of these two types of enzymes support this idea. For example, the $(\alpha/\alpha)_n$ toroid and antiparallel β-sheet folds of GAG lyases are found in families such as PL8 (hyaluronate lyase and chondroitin lyase), PL12 (heparinase III), PL21 (heparinase II), and PL23 (chondroitin lyase). Alginate lyases in PL15, PL17, and PL39 families also exhibit these folds. Additionally, the β-jelly roll fold of heparinase I in the PL13 family and alginate lyases in PL7 and PL18 families show similar structures. Lastly, the β-helix fold of chondroitinase B and alginate lyase in the PL6 family is another example (*Garron and Cygler, 2014*). Considering that the alginate with simpler structure from brown alga or bacteria might appear earlier, it is possible that the ancestral enzymes originally used alginate as a substrate. However, enzymes with transitional characteristics in activity and structure have not been discovered to support this speculation.

Recently, a new class of GAG lyases belonging to PL35 family was discovered with an ultrabroad substrate spectrum (*Wei et al., 2024*). Unlike the identified GAG lyases (HA-specific lyases, CSases, and Hepases) with quite strict substrate specificities, these novel GAG lyases can degrade three types of uronic acid-containing GAGs (HA, CS, and HS), and thus were named GAGases. More interestingly, one of the eight identified GAGases (GAGase III) can even degrade alginate. Analysis of substrate structure preference represented by GAGase I showed that GAGases selectively act on GAG domains composed of non/6-*O*-/*N*-sulfated hexosamines and D-glucuronic acid, among which GAGase III can also selectively degrade polyM blocks composed of D-mannuronic acids but not polyG blocks composed of L-guluronic acids in alginate. Furthermore, the primary structure alignment and phylogenetic analysis showed that GAGases have considerable degree of similarity with not only GAG lyases from PL15, PL21, and PL33 families but also alginate lyases from PL15, PL17, and PL34 families (*Wei et al., 2024*), indicating that GAGases may exhibit some transitional features from alginate lyases to GAG lyases in structure.

In this study, crystal structures of two GAGases (GAGase II and GAGase VII) were determined for the first time, and their catalytic mechanism was further elucidated through the site-directed mutagenesis of their crucial site residues and molecular docking. Structural alignment showed that GAGases are more structural similarity to some alginate lyases rather than GAG lyases, suggesting that PL35 family GAGases might originate from alginate lyases possessing an N-terminal $(\alpha/\alpha)_n$ toroid domain and a C-terminal antiparallel β-sheet domain. Moreover, one of the reasons why GAGase III can additionally degrade M-block alginate are also resolved by structural modeling, structural alignment, and site-directed mutagenesis. This study not only helps to resolve the catalytic mechanism of the PL35 family lyases in particular GAGases, but also provides potential structural evidence for the divergent evolution of GAG lyases.

## Results

### Overall crystal structures of GAGase II and GAGase VII

To explore the substrate recognition and catalytic mechanism of the GAGases, the eight enzymes with different sequence identity (*Supplementary file 1a*) were individually used to prepare their crystals for structural analysis. The structures of GAGase II from *Spirosoma fluviale* (GenBank accession number: SOD82962.1, 81.2% sequence identity with GAGase I), GAGase VII from *Bacteroides intestinalis* (GenBank accession number: EDV05210.1, 44.8% sequence identity with GAGase I) and selenomethionine (SeMet)-labeled GAGase II (Se-GAGase II) in their ligand-free states were solved at 1.9 Å, 2.4 Å, and 2.0 Å resolution, respectively (*Table 1*). Both GAGase II and GAGase VII are monomers consisting of two domains: an N-terminal $(\alpha/\alpha)_6$ toroid domain and a C-terminal two-layered β-sheet domain (*Figure 1A*).

The N-terminal domain (Met[34]-Ala[365]) of GAGase II is composed of 16 large or small α-helices (α1-α16) to form an $(\alpha/\alpha)_6$ toroid domain. Specifically, the N-terminal 16 α-helices constitute six

**Table 1.** Data collection and refinement statistics.

| | GAGase II | GAGase VII | SeMet-GAGase II |
|---|---|---|---|
| PDB entry code | 8KHV | 8KHW | - |
| **Data collection** | | | |
| Wavelength (Å) | 0.98 | 0.98 | 0.98 |
| Resolution range | 44.01–1.90 (1.94–1.90) | 19.87–2.40 (2.49–2.40) | 40.91–2.00 (2.07–2.00) |
| Space group | $P2_1$ | $P4_32_12$ | $P2_1$ |
| Unit cell | a=45.92 Å<br>b=82.10 Å<br>c=79.89 Å<br>$\alpha$=90°<br>$\beta$=106.59°<br>$\gamma$=90° | a=98.41 Å<br>b=98.41 Å<br>c=134.75 Å<br>$\alpha$=90°<br>$\beta$=90°<br>$\gamma$=90° | a=46.39 Å<br>b=81.82 Å<br>c=79.66 Å<br>$\alpha$=90°<br>$\beta$=106.88°<br>$\gamma$=90° |
| Unique reflections | 86421 (4092) | 49305 (2737) | 38212 (3738) |
| Completeness (%) | 98.55 (92.04) | 99.68 (99.85) | 99.15 (98.47) |
| $R_{meas}$ | 0.10 (0.23) | 0.10 (0.74) | 0.08 (0.24) |
| $R_{p.i.m}$ | 0.04 (0.10) | 0.02 (0.16) | 0.03 (0.093) |
| Mean $I/\sigma(I)$ | 12.9 (6.1) | 20.6 (4.3) | 17.9 (8.2) |
| $CC_{1/2}$ | 0.99 (0.96) | 0.99 (0.97) | 0.99 (0.97) |
| **Refinement statistics** | | | |
| $R_{work}$ | 0.16 (0.17) | 0.2079 (0.26) | - |
| $R_{free}$ | 0.1919 (0.23) | 0.2722 (0.36) | - |
| RMSD bond length (A°) | 0.006 | 0.008 | - |
| RMSD bond angle (°) | 0.79 | 0.99 | - |
| Ramachandran plot (%) | | | |
| Favored | 97.75 | 94.31 | - |
| Allowed | 2.25 | 5.00 | - |
| Outliers | 0.00 | 0.69 | - |
| Rotamer outliers (%) | 0.00 | 0.40 | - |
| Clashscore | 2.78 | 6.89 | - |
| Average B-factor (Å²) | 13.51 | 54.14 | - |
| Macromolecules | 11.66 | 54.20 | - |
| Ligands | 21.71 | 45.54 | - |
| Solvent | 25.75 | 47.24 | - |

Statistics for the highest-resolution shell are shown in parentheses.

complete α-helix pairs (α3–4, α5–7, α8–9, α10–12, α13–14, and α16–2), forming six hairpin structures, which eventually embrace into the (α/α)₆ toroid structure. The transition between α6 to α8 and between α10–12 is discontinuous, and both transitions possess a small α-helix α7 and α11, respectively. In each hairpin structure, two antiparallel α-helices are connected by loops composed of 9–12 residues, and each hairpin contains a turn structure composed of 1–4 residues. The N-terminal of this toroid domain is connected to a signal peptide that has been removed during cloning, and its C-terminal is connected to the two-layered β-sheet domain through a 17-residue linkage (Trp³⁵⁰-Met³⁶⁵). Its inner layer is formed by helices α3, α5, α8, α10, α13, and α16, while its outer layer is formed by α2, α4, α6+α7, α9, α11+α12, α14, and α15. Moreover, the α-helix pairs α13–14 and α15–16 are significantly

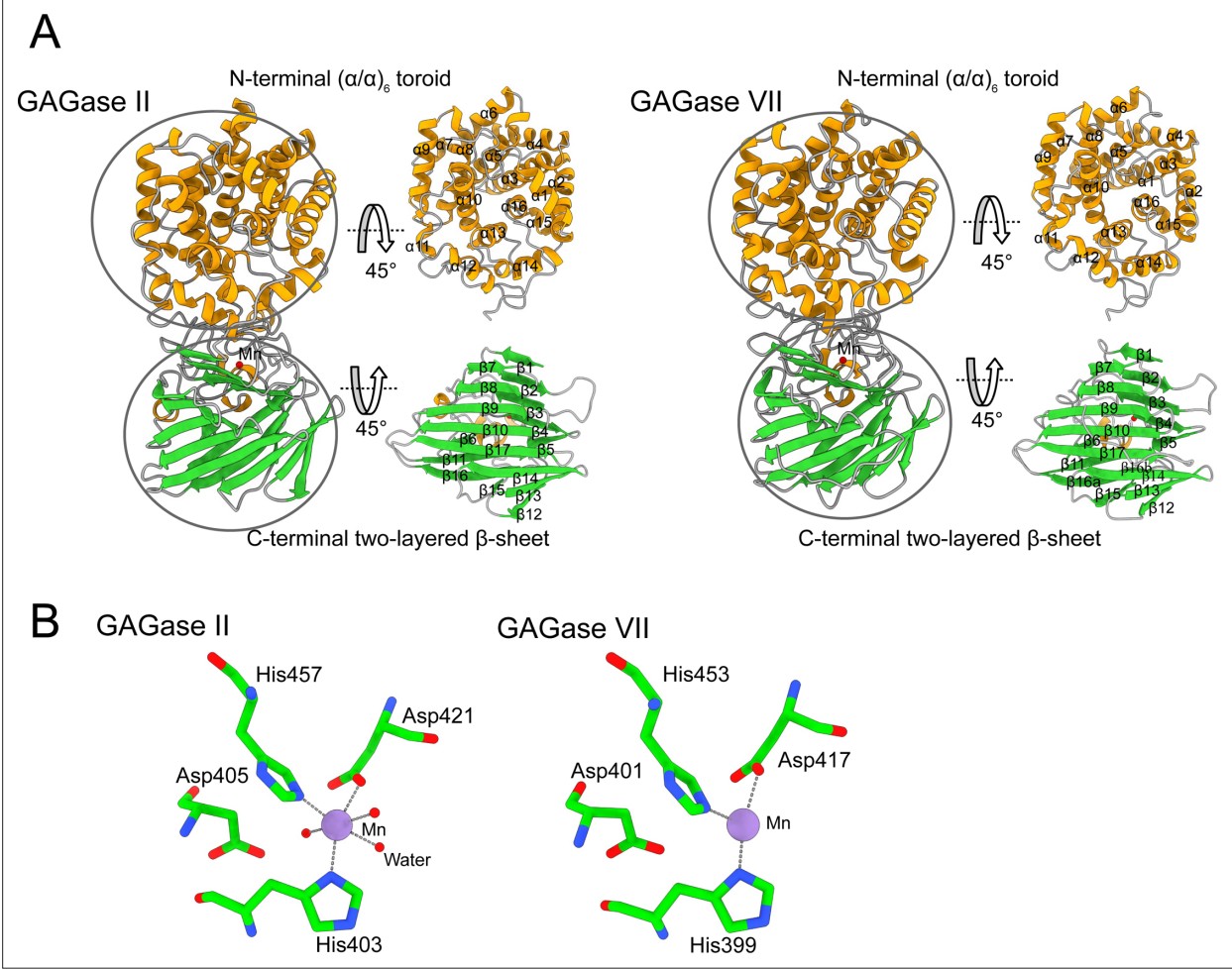

**Figure 1.** Structural description of glycosaminoglycan lyases (GAGase) II and GAGase VII. (**A**) Overall structures of GAGase II (*left*) and GAGase VII (*right*) are shown in carton. The α-helix, β-strand, and random coil are colored with yellow, green, and gray, respectively. The N-terminal (α/α)$_6$ toroid domain and C-terminal two-layered β-sheet domain were circled and the secondary structure elements are marked nearby. (**B**) Mn binding site of GAGase II and GAGase VII. The *purple* sphere presents Mn$^{2+}$. Mn$^{2+}$ binding site of GAGase II (*left*) and GAGase VII (*right*) is shown in stick. A cut-off distance of 3.0 Å was carried out to choose neighboring residues.

The online version of this article includes the following figure supplement(s) for figure 1:

**Figure supplement 1.** Structural modeling and alignment of other glycosaminoglycan lyases (GAGases).

shorter than the first five sets, and α15 and α16 are connected by a short β-turn consisting of only three residues, which makes the toroid structure closed (*Figure 1A*).

The C-terminal domain (Met$^{366}$-Ser$^{612}$) of GAGase II is composed of 17 β-strands (β1-β17) to form a two-layered β-sheet domain with four small α-helices located between β5-β6 and β8-β9. Both layers of this C-terminal β-sheet domain are composed of eight groups of β-strands, namely β1-β5, β11 and β15-β16 and β7-β10, β12-β14, and β17. The two layers are roughly parallel to each other and twisted to form an angle of approximately 60°. The first five β-strands (β1-β5) of the C-terminal domain are arranged in anti-parallel configuration. Subsequently, there is an Ω-loop structure wrapping around a metal ion (Mn$^{2+}$), which consists of three small α-helices and small β-strands, with many residues (such as His$^{401}$, His$^{403}$, Tyr$^{427}$, Glu$^{431}$, Trp$^{438}$, Arg$^{446}$, and so on) on top, which might serve as potential substrate-binding sites. Finally, the C-terminal domain terminates at β17 in the second layer of the bilayer structure. Taken together, these structural elements constitute the C-terminal anti-parallel bilayer β-sheet domain of GAGase II (*Figure 1A*).

Similar to GAGase II, GAGase VII also has an N-terminal (α/α)$_6$ toroid domain (Leu$^{35}$-Arg$^{360}$) composed of 16 α-helices, but its α1 helix consists of only three residues (Glu$^{42}$-Val$^{44}$) and a long loop chain (*Figure 1A*). Moreover, the C-terminal two-layered β-sheet domain (Met$^{361}$-Met$^{616}$) of GAGase

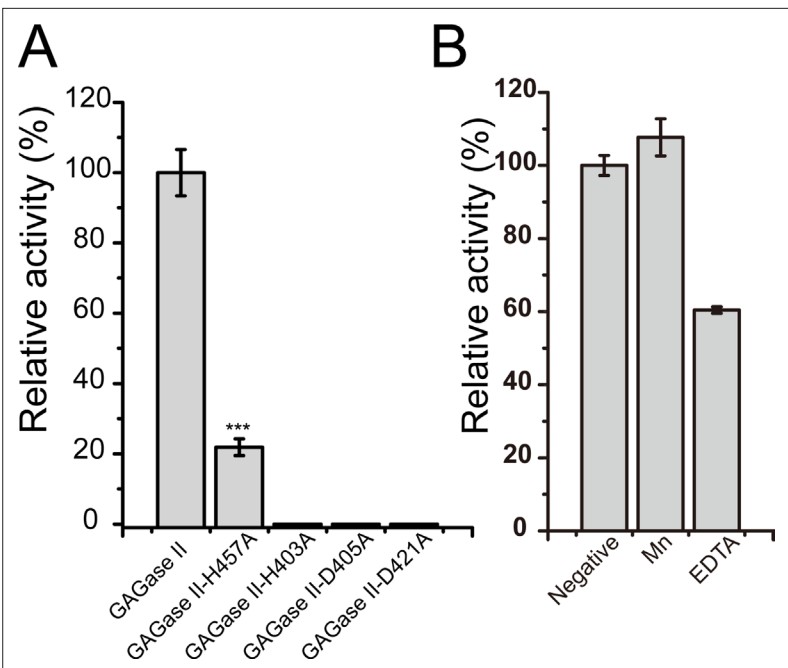

**Figure 2.** Activity of glycosaminoglycan lyases (GAGase II) and its variants. (**A**) Activity of site-directed mutants of His and Asp nearby the $Mn^{2+}$. His and Asp residues nearby the $Mn^{2+}$ were individually mutated to Ala and the relative activity of each variant was measured as described in 'Materials and methods.' All of the residual activities were evaluated and shown as the relative intensity compared with that of wild-type GAGase II. (**B**) The effects of $Mn^{2+}$ and chelating reagent of GAGase II were determined using hyaluronan (HA) (1 mg/ml) as substrate. Error bars represent averages of triplicates (n=3) ± S.D. by Student's t test; ***p<0.001.

VII is also composed of 17 β-strands (β1-β15, β16a, β16b, and β17). Nevertheless, structural alignment further shows that GAGase II has good superposition with GAGase VII and other GAGases, indicating that GAGases are highly conserved in structures (*Figure 1—figure supplement 1*).

## Mn binding site of GAGase II and GAGase VII

Unlike the GAG lyases identified from PL8, PL15, and PL21 families, which contain $Ca^{2+}$ or $Zn^{2+}$ in their overall crystal structures (*Shaya et al., 2008b*; *Shaya et al., 2006*; *Zhang et al., 2021*), a $Mn^{2+}$ was detected in C-terminal domains of both GAGase II and VII by inductively coupled plasma-mass spectrometry (ICP-MS) analysis (*Figure 1B*, *Supplementary file 1b*). Structurally, the $Mn^{2+}$ is located inside the long Ω loop after the β1-β5 antiparallel β-sheet, and coordinates with the nearby nitrogen and oxygen atoms of several residues such as Asp and His in both GAGase VII and GAGase II (GAGase II also has coordination with three surrounding water molecules), with distances of 2.0–2.5 Å. Such interaction should stabilize the nearby loop structures and may form a substrate-binding site relevant for substrate selectivity. To investigate the roles of these adjacent residues interacting with the $Mn^{2+}$, $His^{403}$, $Asp^{405}$, $Asp^{421}$, and $His^{457}$ in GAGase II were individually mutated to alanine (Ala), and the results showed that these variants resulted in complete or substantial inactivation of GAGase II (*Figure 2A*). In addition, the inhibitory effects of the chelator (EDTA) and the stimulation of $Mn^{2+}$ on the enzymatic activities also support the critical role of $Mn^{2+}$ on GAGases (*Figure 2B*).

## Multiple structural alignments of GAGase II and its structurally similar GAG lyases and alginate lyases

The ultrabroad substrate degradation spectra of GAGases indicate that they should be structurally similar with some other PLs with different substrate specificities. By using GAGase II as a representative, a structural alignment assay showed that the PL35 GAGases are structurally similar to various N-terminal $(\alpha/\alpha)_n$ toroid domain and C-terminal antiparallel β-sheet domain-containing GAG/alginate lyases from PL8 (*Féthière et al., 1999*; *Huang et al., 2003*; *Li and Jedrzejas, 2001*; *Lunin et al., 2004*; *Shaya et al., 2008a*), PL12 (*Hashimoto et al., 2014*; *Ulaganathan et al., 2017*), PL15 (*Ochiai*

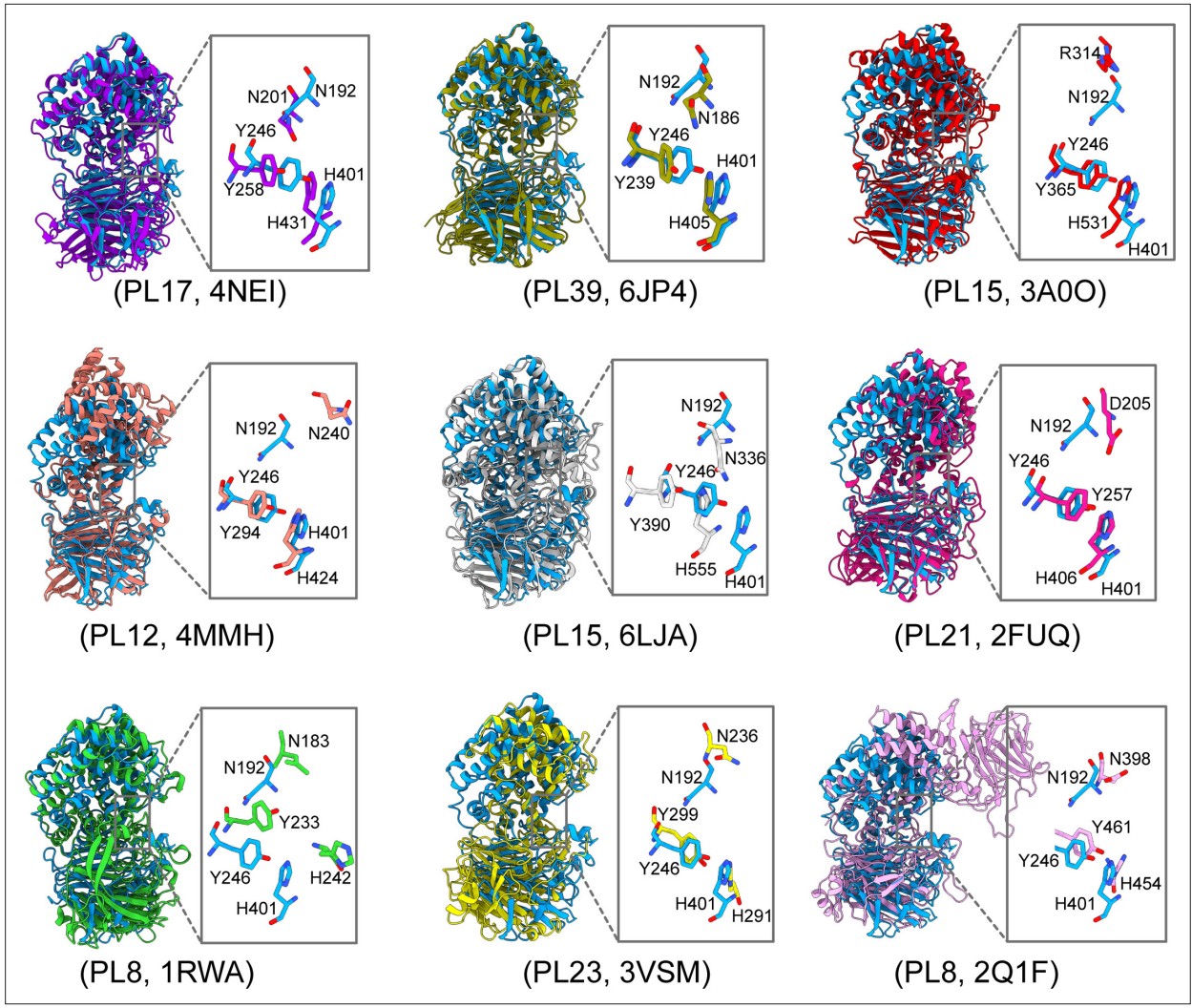

**Figure 3.** Multiple structural alignments of glycosaminoglycan lyases (GAGase) II and its structurally similar proteins. GAGase II (8KHV, *blue*) was aligned with structurally identified glycosaminoglycans (GAGs) and alginate lyases, including PL17 family alginate lyase (4NEI, *purple*), PL39 family alginate lyase (6JP4, *olive*), PL15 family alginate lyase (3A0O, *red*), PL12 family heparinase III (4MMH, *salmon*), PL15 family exoHep (6LJA, *gray*), PL21 family heparinase II (2FUQ, *magenta*), PL8 family chondroitin sulfate AC lyase II (1RWA, *green*), PL23 family chondroitinase (3VSM, *yellow*), and PL8 family chondroitin sulfate ABC lyase II (2Q1F, *pink*). The detailed views of the crucial active site residues are shown in stick mode. The root-mean-square deviation (RMSD) between these structures and GAGase II were calculated as 1.34, 1.34, 1.30, 1.24, 1.15, 1.35, 1.24, 1.14, and 1.41 Å based on 171, 148, 141, 111, 99, 87, 60, 22, and 21 pruned atoms, respectively.

*et al., 2010*; *Zhang et al., 2021*), PL17 (*Park et al., 2014*), PL21 (*Shaya et al., 2006*), PL23 (*Sugiura et al., 2011*), and PL39 (*Ji et al., 2019*). Notably, GAGase II shares very low sequence identity with PL21 heparinase II from *Pedobacter heparinus* (*Shaya et al., 2006*) (23.1% identity with 29% query coverage), PL12 heparinase III from *Bacteroides thetaiotaomicronin* (*Ulaganathan et al., 2017*) (23.6% identity with 32% query cover) and PL39 alginate lyase from *Defluviitalea phaphyphila* (*Ji et al., 2019*) (26.5% identity with 31% coverage), and no significant sequence identity with PL17 alginate lyases (*Park et al., 2014*), PL15 exoHep (*Zhang et al., 2021*), PL8 chondroitin ABC lyase (*Shaya et al., 2008a*) and chondroitin AC lyase *Lunin et al., 2004*; however, all are relatively well superimposable in their overall structures and highly conserved in their triplet active site residues (*Figure 3*). In terms of overall structural similarities, the (α/α)n toroid domain and anti-parallel β-sheet domain of GAGase II show more structural similarity to alginate lyases with (α/α)6 toroid and the antiparallel β-sheet domain, such as those from the PL15 (PDB code: 3A0O) (*Ochiai et al., 2010*), PL17 (PDB code: 4NEI) (*Park et al., 2014*), and PL39 (PDB code: 6JP4) (*Ji et al., 2019*) families, than various GAG lyases (*Tables 2 and 3*), indicating that GAGases might have originated from alginate lyases rather than GAG

**Table 2.** Structural similarity of glycosaminoglycan lyases (GAGase) II/GAGase VII with glycosaminoglycan (GAG)/alginate lyases analyzed using DALI.

| | Chain | Z-score | RMSD (Å) | lali | nres | Description | PL family |
|---|---|---|---|---|---|---|---|
| | 3a0o-A | 29.4 | 3.2 | 539 | 764 | Alginate lyase | 15 |
| | 6jp4-A | 28.9 | 3.0 | 523 | 770 | Alginate lyase | 39 |
| | 4nei-A | 27.3 | 3.1 | 523 | 705 | Alginate lyase | 17 |
| | 4mmh-A | 28.6 | 4.4 | 481 | 637 | Heparinase III | 12 |
| | 6lja-A | 25.7 | 3.9 | 540 | 841 | Exo-type heparinase | 15 |
| | 2fuq-A | 25.1 | 4.9 | 528 | 743 | Heparinase II | 21 |
| | 1rwa-A | 15.2 | 5.4 | 470 | 754 | Chondroitin AC lyase | 8 |
| | 3vsm-a | 14.3 | 5.3 | 369 | 633 | Chondroitinase | 23 |
| GAGase II | 2q1f-a | 11.1 | 5.3 | 434 | 991 | Chondroitin ABC lyase | 8 |
| | 3a0o-A | 29.2 | 3.6 | 545 | 764 | Alginate lyase | 15 |
| | 6jp4-A | 28.0 | 3.3 | 530 | 770 | Alginate lyase | 39 |
| | 4nei-A | 27.2 | 3.2 | 529 | 705 | Alginate lyase | 17 |
| | 4mmh-A | 26.4 | 4.4 | 492 | 637 | Heparinase III | 12 |
| | 2fuq-A | 23.1 | 5.4 | 531 | 747 | Heparinase II | 21 |
| | 1rwa-A | 14.3 | 4.5 | 473 | 754 | Chondroitin AC lyase | 8 |
| | 3vsm | 12.9 | 4.3 | 365 | 633 | Chondroitinase | 23 |
| GAGase VII | 2q1f-A | 10.5 | 4.6 | 407 | 991 | Chondroitin AC lyase | 8 |

**Table 3.** Multiple structural alignments of $(\alpha/\alpha)_n$ toroid domain or antiparallel β-sheet domain of glycosaminoglycan lyases (GAGase) II and identified glycosaminoglycan (GAG)/alginate lyases.

| | | PL family | Domain | Aligned atoms | RMSD | Substrate |
|---|---|---|---|---|---|---|
| | 8KHV | PL35 | $(\alpha/\alpha)_6$ toroid (Met$^{34}$-Trp$^{350}$) | 317 | 0.00 | HA, CS, and HS |
| | 3A0O | PL15 | $(\alpha/\alpha)_6$ toroid (Gly$^{127}$-Leu$^{452}$) | 101 | 1.04 | Alginate |
| | 4NEI | PL17 | $(\alpha/\alpha)_6$ toroid (His$^{30}$-Glu$^{364}$) | 102 | 1.20 | Alginate |
| | 6JP4 | PL39 | $(\alpha/\alpha)_6$ toroid (Ala$^{1}$-Tyr$^{355}$) | 113 | 1.30 | Alginate |
| | 2FUQ | PL21 | $(\alpha/\alpha)_6$ toroid (Thr$^{27}$-Leu$^{368}$) | 67 | 1.42 | Hep, HS |
| | 1RWA | PL8 | $(\alpha/\alpha)_6$ toroid (Pro$^{4}$-Val$^{367}$) | 60 | 1.24 | HA, CS |
| | 3VSM | PL23 | $(\alpha/\alpha)_5$ toroid (Asn$^{72}$- Asn$^{330}$) | 22 | 1.14 | CS |
| | 7FHU | PL5 | $(\alpha/\alpha)_6$ barrel (Cys$^{23}$-Pro$^{329}$) | 27 | 1.105 | Alginate |
| | 8BDQ | PL38 | $(\alpha/\alpha)_7$ barrel (Ala$^{24}$-Lys$^{402}$) | 58 | 1.177 | Alginate |
| $(\alpha/\alpha)_n$ toroid domain | 4MMH | PL12 | $(\alpha/\alpha)_5$ toroid (Ile$^{29}$-Thr$^{379}$) | 24 | 1.169 | Hep, HS |
| | 8KHV | PL35 | anti-parallel β-sheet | 261 | 0.00 | HA, CS, and HS |
| | 3A0O | PL15 | anti-parallel β-sheet | 109 | 1.11 | Alginate |
| | 4NEI | PL17 | anti-parallel β-sheet | 85 | 0.97 | Alginate |
| | 6JP4 | PL39 | anti-parallel β-sheet | 67 | 1.06 | Alginate |
| antiparallel β-sheet domain | 2FUQ | PL21 | anti-parallel β-sheet | 103 | 1.17 | Hep, HS |
| | 4MMH | PL12 | anti-parallel β-sheet | 97 | 1.22 | Hep, HS |

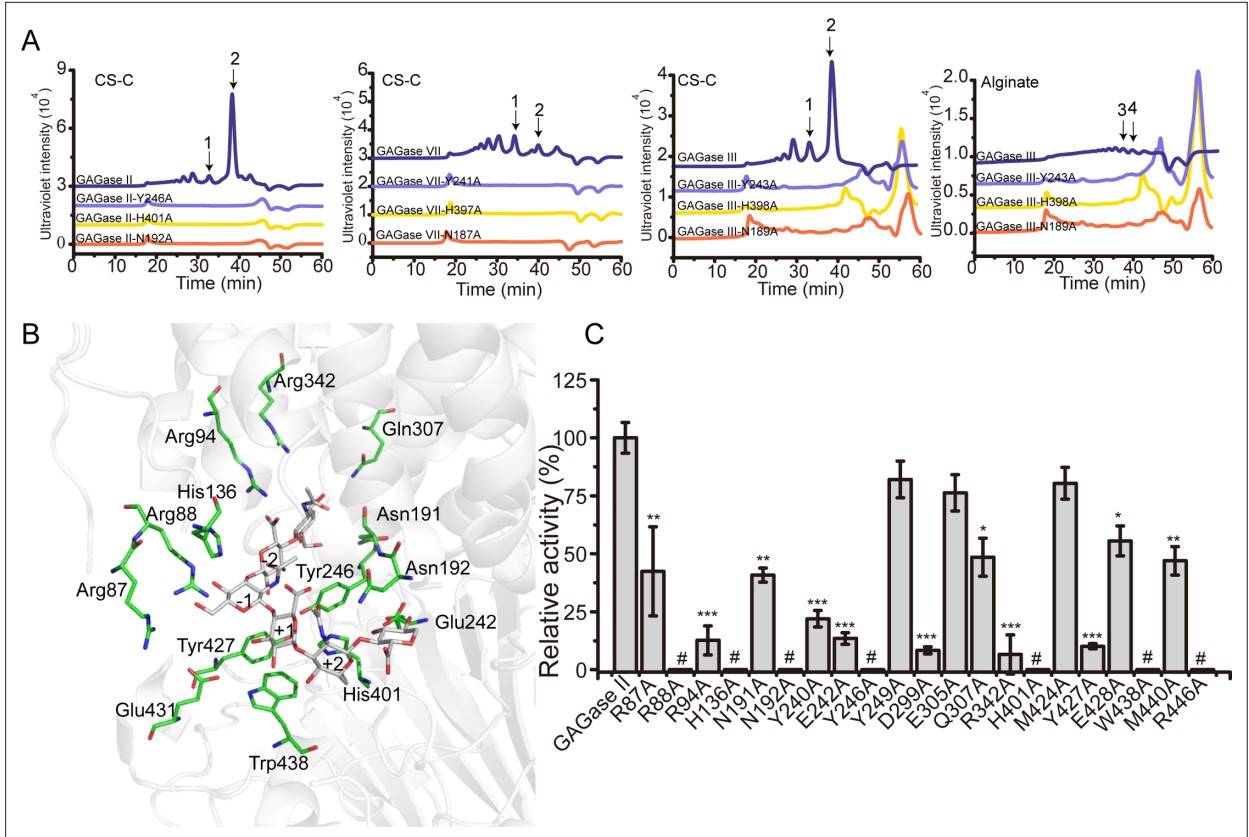

**Figure 4.** Catalytic center and substrate binding sites of glycosaminoglycan lyases (GAGases). (**A**) The crucial catalytic site-directed mutagenesis of GAGase II, GAGase III, and GAGase VII. The conserved crucial residues of GAGase II, GAGase III, and GAGase VII were individually mutated to Ala. CS-C and alginate were used as substrates for the activity evaluation of GAGase II, GAGase III, GAGase VII, and its variants. The activity of each variant was detected using gel filtration HPLC on a Superdex Peptide column as described in 'Materials and methods;' the elution of each fraction is indicated as follows: 1, CS-C tetrasaccharide; 2, CS-C disaccharide; 3, alginate trisaccharide; 4, alginate disaccharide. (**B**) Molecular docking of GAGase II with a hyaluronan (HA) hexasaccharide. The molecule docking was carried out with GAGase II and a HA hexasaccharide (PDB code: 1HYA) to predict the substrate binding sites. The binding site residues (*green*) and hexasaccharide ligand (*gray*) are showed as sticks. (**C**) The putative substrate-binding site residues surrounding the docking substrate were individually mutated to Ala. HA was treated with each variant at 40 °C for 12 hr and the relative activity of each variant was shown as the relative intensity compared with that of wild-type GAGase II. Error bars represent averages of triplicates (n=3) ± S.D. by Student's t test; *p<0.01, **p<0.001, ***p<0.0001, #: the activities of the variants were too low to be accurately detected and almost completely inactivated.

The online version of this article includes the following source data and figure supplement(s) for figure 4:

**Figure supplement 1.** Surface representations and substrate binding tunnels of glycosaminoglycan lyases (GAGase) II and its structurally similar glycosaminoglycan (GAG)/alginate lyase.

**Figure supplement 2.** SDS-PAGE of glycosaminoglycan lyases (GAGases) and their variants.

**Figure supplement 2—source data 1.** The original files of the full raw uncropped, unedited gels.

**Figure supplement 2—source data 2.** Figures with the uncropped gels or blots with the relevant bands clearly labeled.

lyases. Reasonable support for this speculation could be the significant alginate-degrading activity of GAGase III. Therefore, the broad substrate spectrum of GAGases may be because they are evolutionarily transitional types with the enzymatic features of alginate lyases and GAG lyases.

## Catalytic center and substrate binding sites of GAGases

Based on the sequence and structural alignment with homogenous alginate/GAG lyases, the conserved residues Tyr[246], His[401], Asn[192] of GAGase II, Tyr[243], His[398], Asn[189] of GAGase III, and Tyr[241], His[397], Asn[187] of GAGase VII may be the key triplet residues involved in β-elimination catalysis (*Figure 3*). To confirm this hypothesis, these residues were individually replaced by Ala and the activity of the mutants was analyzed. The results showed that the activity of all variants toward CS-C was completely lost

**Table 4.** Catalytic residues comparison of identified glycosaminoglycans (GAGs) and alginate lyases shared similar structure with glycosaminoglycan lyases (GAGase) II and GAGase VII.

| | PDB code | PL family | Substrates | Sequence identity (Query cover/Per. Ident) * | Catalytic residues | |
| --- | --- | --- | --- | --- | --- | --- |
| | | | | | General base/acid | Neutralizer |
| GAGase II | 8KHV | PL35 | HA, CS, and HS | 100%/100% | Tyr246, His401 | Asn192 |
| GAGase VII | 8KHW | PL35 | HA, CS, and HS | 93%/44.67% | Tyr241, His397 | Asn187 |
| GAGase III | - | PL35 | HA, CS, HS, and alginate (M-specific) | 98%/64.08% | Tyr243, His398, | Asn189, His188 |
| Chondroitin sulfate AC lyase II | 1RWA | PL8 | HA and CS | N.D.† | His233, Tyr242 | Asn183 |
| Chondroitin sulfate ABC lyase II | 2Q1F | PL8 | HA, CS, and DS | N.D. | His345, Tyr461, His454 | Asp398 |
| Heparinase III | 4MMH | PL12 | HS | 8%/30.91% | Tyr294, His424 | Asn240 |
| Alginate lyase | 3A0O | PL15 | Alginate (M/G specific) | 43%/23.93% | His311, Tyr365, His531 | Arg314 |
| Exo-Heparinase | 6LJA | PL15 | Hep and HS | N.D. | His337, Tyr390, His555 | Asn336 |
| Alginate lyase | 4NEI | PL17 | Alginate (M-specific) | 15%/25.51% | Tyr258, His413 | Asn201, His200 |
| Heparinase II | 2FUQ | PL21 | Hep and HS | 29%/23.13% | His202, Tyr257, His406 | Glu205 |
| Chondroitin lyase | 3VSM | PL23 | CS | N.D. | Tyr299, His291 | Asn236 |
| Alginate lyase | 6JP4 | PL39 | Alginate (M/G specific) | 31%/26.53% | Tyr239, His405 | Asn186, His187 |

*Sequence identity means the sequence similarity compared with GAGase II.
†N.D. means no significant identity is detected.

(*Figure 4A*), indicating that GAGases act through a general acid-base catalytic mechanism by using His/Tyr as a Brønsted acid/base, similar to that observed for various other identified GAG/alginate lyases (*Table 4*). Compared with structurally similar GAG/alginate lyases, the results from molecular docking and catalytic tunnel predictions showed that GAGase II contains a shorter catalytic cavity (*Figure 4—figure supplement 1*), which may explain why they can accommodate and degrade a variety of substrates with very different structures. However, we were unsuccessful in preparing the cocrystals of GAGases/their inactive variants with substrates to reveal their substrate binding and degradation mechanisms. Alternatively, molecular docking experiments of GAGase II with several structurally defined hexasaccharide (Hexa) substrates were performed as shown in *Figure 4B*. Based on the docking results obtained for GAGase II with an HA Hexa $(GlcUA1-3GlcNAc)_3$ (PDB code: 1HYA) (*Winter et al., 1975*; *Figure 4B*), the nonreducing end of the substrate at the '−1' and '−2' subsites is near some positively charged amino acids, including Arg[87], Arg[88], Arg[94], His[136], Asn[191] and Asn[192], in the vicinity of the catalytic cavity of GAGase II, which could facilitate the binding of negatively charged substrates. Besides, several acidic amino acids (Glu[242], Asp[299], and Glu[431]) are very close to the '+1' and '+2' subsites, which could promote the release of the reducing end product via charge repulsion. There are some other residues surrounding around the catalytic cavity, such as Tyr[240], Gln[307], Arg[342], Tyr[427], Glu[428], Glu[431], Trp[438], Met[440], and Arg[446], which may play a key role in stabilization of the catalytic cavity and overall structure. Site-directed mutagenesis of these residues individually replaced with Ala showed that resulting variants were partially or completely inactivated (*Figure 4C*), confirming that these residues play important roles in the binding and release of substrate and structural stabilization.

## Substrate selectivity of GAGases

As mentioned above, GAGase III can degrade not only HA, CS, and HS but also alginate, even though the activity on alginate is relatively low (about 1/50 of the activity on HA). To investigate the reason why this enzyme is capable to act on alginate, its structure was predicted using RoseTTAfold, and aligned with GAGase II and GAGase VII. Results showed that GAGase III has a unique His[188] residue in

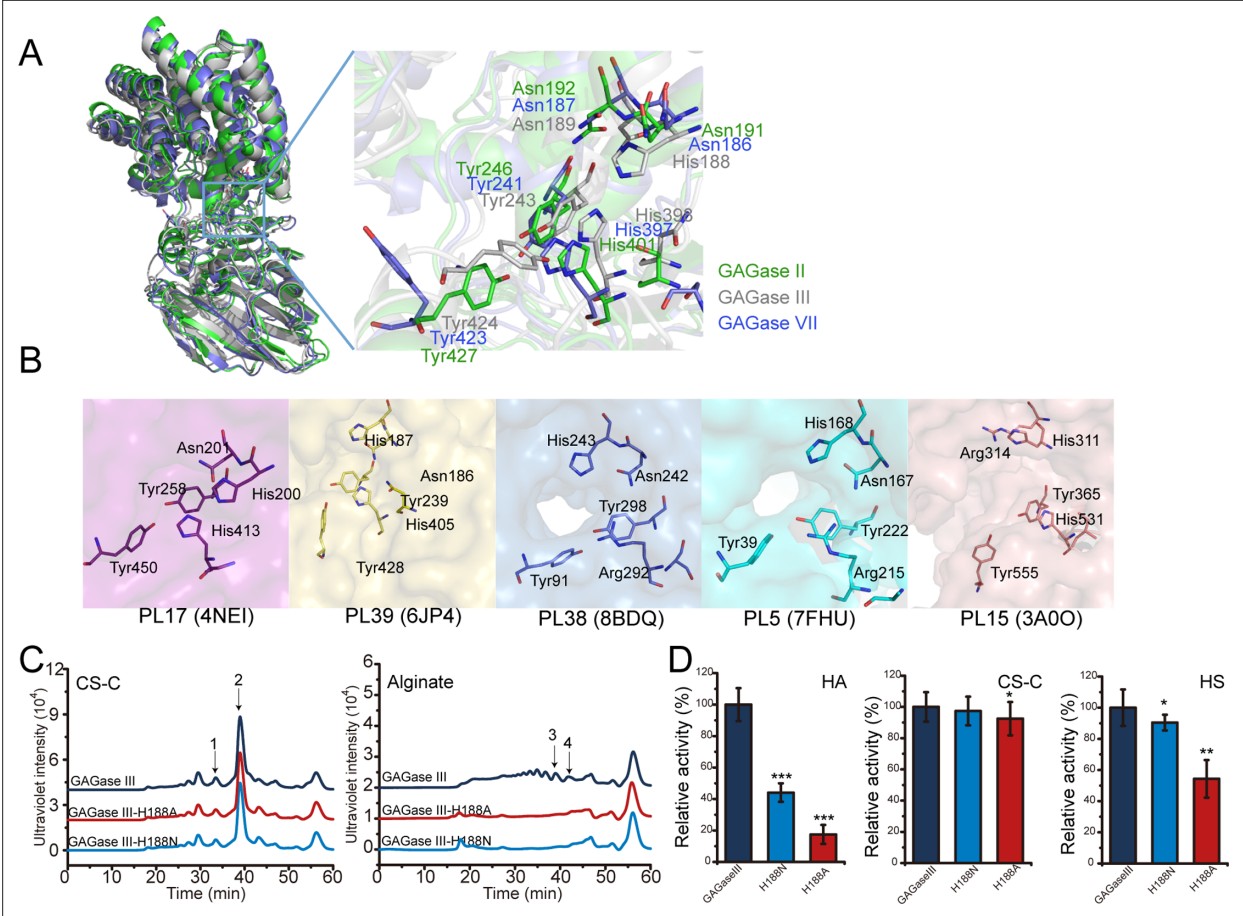

**Figure 5.** Analysis of a key residue for the alginate-degrading activity of glycosaminoglycan lyases (GAGase) III. (**A**), Multiple structural alignment of GAGase II, III, and VII. GAGase II (8KHV, *green*) and GAGase VII (8KHW, *skyblue*) were aligned with GAGase III (model, *gray*). The detailed views of crucial catalytic residues are shown in stick mode; (**B**), Conserved catalytic residues of alginate lyases from PL17 (4NEI, *purple*), PL39 (6JP4, *yellow*), PL38 (8BDQ, *slate*), PL5 (7FHU, *cyan*) and PL15 (3A0O, *pink*) family. Residues are shown in stick; (**C**), Activity assay of GAGase III-H188N and GAGase III-H188A toward CS-C and alginate. The crucial site His[188] was mutated to alanine and asparagine, respectively. The activity of each variant against CS-C and alginate was assessed using gel filtration HPLC on a Superdex Peptide column, as described under 'Materials and methods;' the elution of each fraction is indicated as follows: 1, CS-C tetrasaccharide; 2, CS-C disaccharide; 3, alginate trisaccharide; 4, alginate disaccharide; (**D**), Relative activity of GAGase III and its variants. Three types of GAGs (HA, CS-C, and HS) were individually treated with GAGase III and its variants (GAGase III-H188N and GAGase III-H188A) at 40 °C for 1 hr; relative activities of enzymes were determined by detecting the absorbance at 232 nm. Data are shown as the percentage of the activity relative to the wild-type GAGase III. Error bars represent averages of triplicates (n=3) ± S.D. by Student's t test; *p<0.5; **p<0.01; ***p<0.001.

The online version of this article includes the following figure supplement(s) for figure 5:

**Figure supplement 1.** Activity assay of glycosaminoglycan lyases (GAGases) variants against CS-C and alginate.

its catalytic cavity, which is conserved in alginate lyases from PL17 (*Park et al., 2014*), PL39 (*Ji et al., 2019*), PL38 (*Rønne et al., 2023*), PL5 (*Pandey et al., 2021*), and PL15 (*Ochiai et al., 2010*) families, while other GAGases have a conserved asparagine residue such as the Asn[191] of GAGase II and the Asn[186] of GAGase VII in the corresponding position (*Figure 5A–B*). To verify the function of this histidine, His[188] was replaced with alanine and asparagine, respectively. The results of substrate-specificity analysis showed that the alginate-degrading activity of both GAGase III-H188A and GAGase III-H188N were abolished even at a quite high ratio of the mutated enzyme to substrate such as 30 µg enzyme to 30 µg substrate (*Figure 5—figure supplement 1A*), while their GAG-degrading activity was only partially affected, indicating that the His[188] residue is essential for the alginate-degrading activity of GAGase III (*Figure 5C–D*). However, when the asparagine residue at the corresponding position of other GAGases was mutated to histidine, the obtained variants did not exhibit any alginate-degrading activity (*Figure 5—figure supplement 1B*), suggesting that degrading activity of GAGase II remains to be determined outside of the His[188] residue.

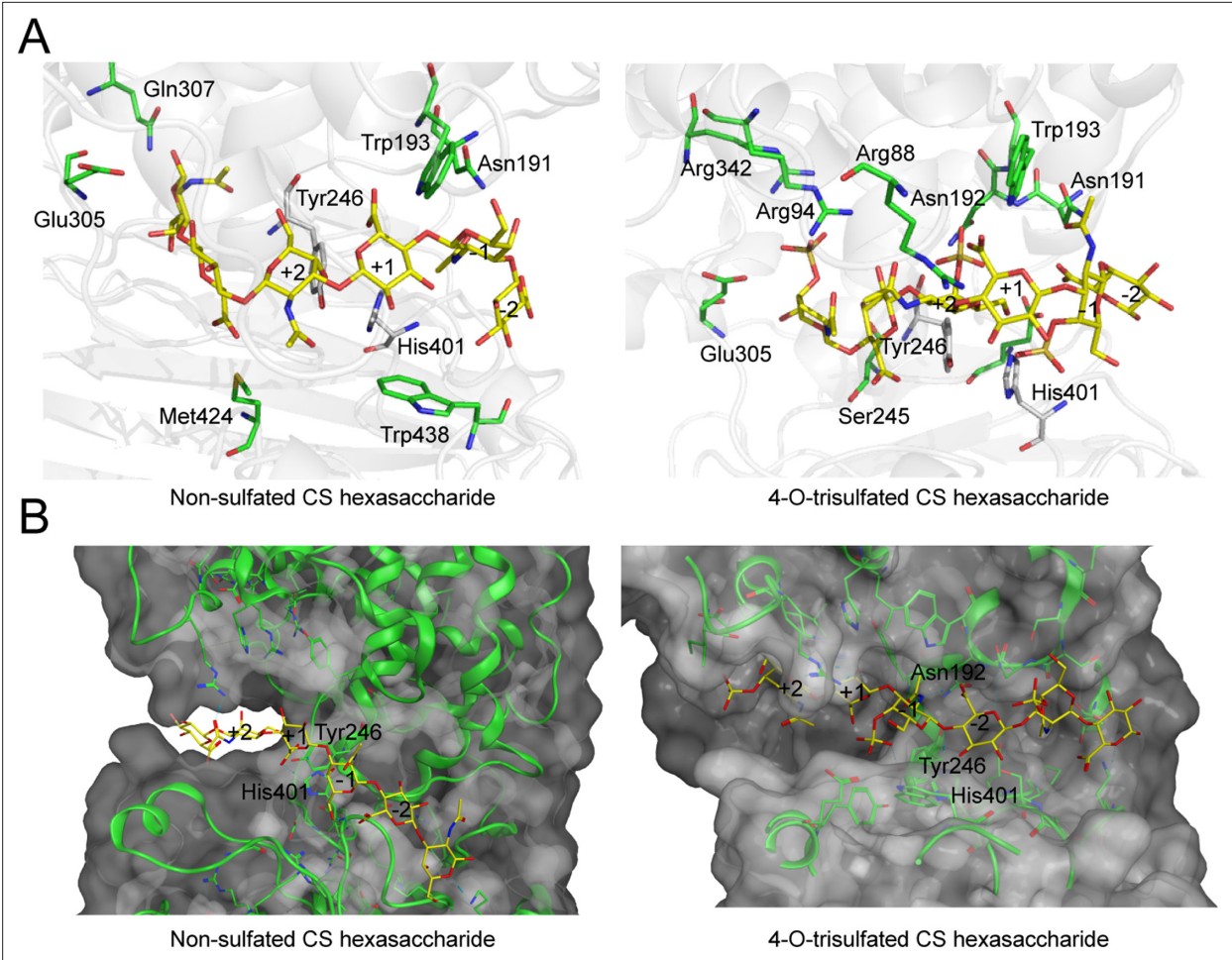

**Figure 6.** Molecular docking of glycosaminoglycan lyases (GAGase) II with hexasaccharide ligands. Molecular docking of GAGase II with chondroitin sulfate (CS) ligands. The molecule docking was carried out with GAGase II and CS ligands, including nonsulfated (PDB code: 2KQO) (*left*) and 4-*O*-trisulfated (PDB code: 1C4S) (*right*) CS hexasaccharide, to investigate the substrate selectivity, using AutoDock Vina (**A**) and Molecular Operating Environment (MOE) (**B**). The catalytic triplet residues (*green*) and CS ligand (*yellow*) are showed as sticks.

The online version of this article includes the following figure supplement(s) for figure 6:

**Figure supplement 1.** wo-dimensional interaction plot of glycosaminoglycan lyases (GAGase) II with molecular docking substrates.

GAGases show preference for some specific sulfation patterns (*Wei et al., 2024*). For example, GAGases prefer to degrade CS domains composed of non-/6-*O*-sulfated but not 4-*O*-sulfated GalNAc residues and D-GlcA residues. To further investigate the reason why these enzymes have such structural preference for substrates, we tried to prepare the co-crystal of GAGase II or VII with various structure-defined oligosaccharide substrates, but ultimately failed. Therefore, we used molecular docking analysis to preliminarily explore the possible reasons why GAGases have substrate structural preference. To this end, docking of GAGase II with a nonsulfated CS Hexa (GlcUAβ1–3GalNAc)$_3$ (PDB code: 2KQO) and a tri-4-*O*-sulfated CS Hexa (GlcUAβ1–3GalNAc(4 S))$_3$ (PDB code: 1C4S) (*Sattelle et al., 2010*) were carried out, respectively. In the docked model, the monosaccharide residues of the substrate were named '+1' and '+2' toward reducing end, and '−1' and '−2' toward non-reducing end from the β1–4 cleavage site following the nomenclature (*Davies et al., 1997*). The results show that for the nonsulfated CS Hexa, the GlcUA residue at the '+1' subsite is closely approached by the Tyr[246] residue as a Brønsted base (*Figure 6 left*), which facilitates the uptake of the C5 proton of GlcUA by Tyr[246], and in the case of tri-4-*O*-sulfated CS Hexa, the 4-*O*-sulfate group in GalNAc residue at the '+2' subsite interacts with the neutralizer Asn[192] and the catalytic residue Tyr[246] via hydrogen bonds etc., the C5 proton of GlcUA at the '+1' subsite is away from the key catalytic residue Tyr[246] (*Figure 6 right*, *Figure 6—figure supplement 1*), which hinders the degradation of CS with such sulfation pattern, as

we detected in the biochemical analysis above. Of course, the results from docking assay need to be further confirmed by more structural and biochemical evidence.

## Catalytic mechanism of GAGases

In summary, taking GAGase II as a representative, the catalytic process of GAGases is proposed as follows. The negatively charged GAG substrate binds to several positively charged residues, such as Arg[87], Arg[88], Arg[94], His[136], Asn[191], and Asn[192], near the catalytic triad residues in the catalytic cavity. The Asn[191] and Asn[192] residues interact with the carboxyl group of GlcUA at the +1 subsite to reduce the p$K$a and promote the protonation of the C5 proton. The His[401] works as a proton receptor (Brønsted base) to abstract protons from the C5 position of GlcUA and create an enolate tautomeric. The protonated Tyr[246] (Brønsted acid) provides a proton to the C-4–O-1 glycosidic bond between the '−1' and '+1' subsite. The glycosidic bond is cleaved as an electron transfer occurs from the carboxyl group to form a double bond between C4 and C5 of GlcUA. A new reducing end and a new unsaturated nonreducing end are recreated. Moreover, the degradation of polyM by GAGase III requires the involvement of His[188] residue, which may be involved in the neutralization of carboxyl groups in M-blocks in alginate (*Figure 7*).

## Discussion

GAG lyases derived from microorganisms, as GAG-specific degrading enzymes, are classified into 13 PL families in the CAZy database (*Drula et al., 2022*). They are not only critical for microorganisms to degrade and utilize GAGs as carbon sources, but also become useful tools for structural and functional studies of GAGs. In contrast, alginate lyases, as alginate-specific degrading enzymes, are classified into 15 PL families in the CAZy database (*Drula et al., 2022*). They are essential for the metabolism of alginate in microorganisms and algae as well as lower marine animals, and have completely different substrate specificities than GAG lyases. Due to the similarity between GAG lyases and alginate lyases in 3D fold, GAG lyases are thought to have originated from the divergent evolution of alginate lyases (*Garron and Cygler, 2014*), but enzymes that exhibit transitional features in function and structure, which would provide direct evidence to support this view, are lacking. On the basis of previous functional studies (*Wei et al., 2024*), structural studies of GAGases with HA, CS, HS, and even alginate-degrading activities should further provide substantial evidence for the evolution of GAG lyases.

Prior to this study, no member of the PL35 family had been structurally characterized. Here, the structures of GAGase II and VII were determined at resolutions of 1.9 Å and 2.4 Å, respectively, and both exhibit a highly similar structure of 'N-terminal (α/α)₆ toroid and C-terminal two-layer antiparallel β-sheet,' which is also the fold adopted by most structurally known GAG lyases except CSase B in PL6 (*Huang et al., 1999*), Hepase I in PL13 (*Han et al., 2009*) and hyaluronan lyase in PL16 (*Martinez-Fleites et al., 2009*). Structural alignment has shown that GAGases share considerable degrees of similarity with all GAG/alginate lyases with the (α/α)ₙ toroid and antiparallel β-sheet fold in PL8 (*Féthière et al., 1999*), PL12 (*Hashimoto et al., 2014*), PL15 (*Ochiai et al., 2010*; *Zhang et al., 2021*), PL17 (*Park et al., 2014*), PL21 (*Shaya et al., 2006*), PL23 (*Sugiura et al., 2011*), and PL39 (*Ji et al., 2019*) families, especially the key catalytic triplet residues (Tyr, His, and Asn) in their active centers, which are highly conserved; thus, they may share a similar Brønsted base/acid catalytic mechanism (*Garron and Cygler, 2010*; *Garron and Cygler, 2014*). Notably, GAGases are more structurally similar to alginate lyases from PL15 (*Ochiai et al., 2010*), PL17 (*Park et al., 2014*), and PL39 (*Ji et al., 2019*) family than to various GAG lyases, which further structurally supports the speculation that GAGases originated from alginate lyases.

As mentioned previously, many different GAG/alginate lyases share similar catalytic sites and the same catalytic machinery (*Garron and Cygler, 2010*; *Garron and Cygler, 2014*), but their differences in substrate selectivity and endo-/exolytic manner are mainly attributed to subtle differences in the three-dimensional structure (*Lombard et al., 2010*), which may be the substrate-binding sites. The recognition of appropriate substrates by PLs mainly depends on the interaction between the positively charged side chains of basic amino acid residues in their substrate-binding sites and the negatively charged carboxyl groups of HexUA residues in substrates. Moreover, compared to the structurally and biochemically characterized members of the PL8, PL12, PL15, PL21, and PL39 families, PL35 family proteins possess a shorter catalytic cavity; thus, PL35 family proteins may more easily

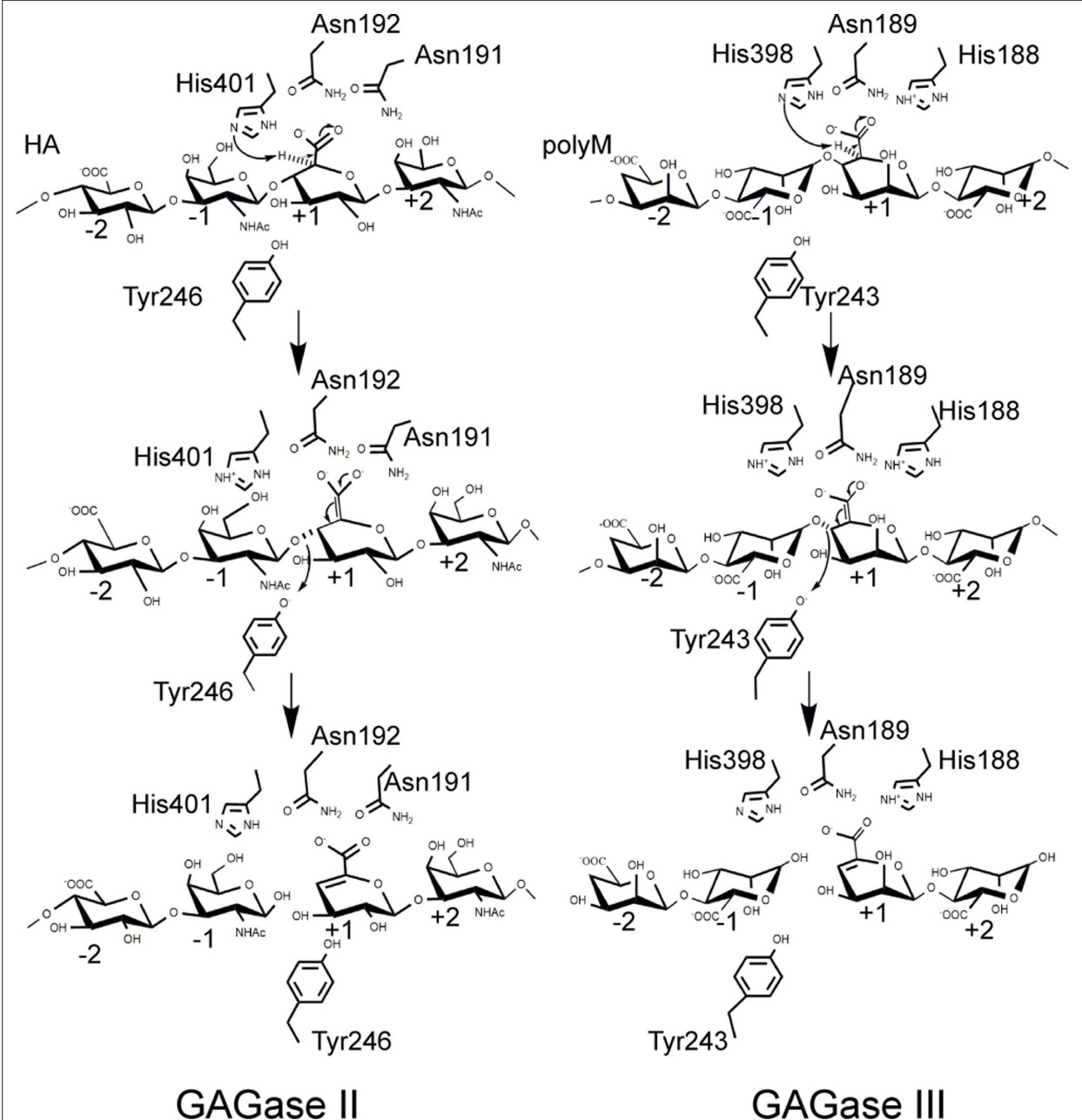

**Figure 7.** Proposed catalytic mechanism of glycosaminoglycan lyases (GAGase) II and GAGase III. Take the hyaluronan (HA) degradation by GAGase II and the polyM degradation by GAGase III for example. Briefly, the substrate is firstly binding to the negatively charged residues near the catalytic sites, the carboxylate group is neutralized by Asn[192], Asn[191] in GAGase II and His[188], Asn[189] in GAGase III; His[398] in GAGase II and His[401] in GAGase III are proposed to work as general base to abstract a proton from the C5 position of GlcUA at +1 position, and Tyr[246] in GAGase II and Tyr[243] in GAGase III work as general acid to donate the leaving group at −1 position a proton. The arrows indicate the direction of electron transfer, and the C5-C6 double bond on the middle panel indicate the enolate anion intermediate created by proton abstraction at C5 position.

accommodate various substrates with different structures but may exhibit a weaker ability to bind and degrade specific substrates, resulting in the activities of GAGases towards various GAGs and alginate are much lower than those of GAG or alginate-specific lyases. Certainly, the accurate substrate selectivity mechanism of GAGases remains to be revealed by preparing and resolving co-crystals of the enzyme-substrate complex in the future.

In addition, GAGases tend to act on GlcUA-containing HA, CS, and HS but not IdoUA-containing DS and Hep, which should result from the absence of a His residue functioning as an *anti*-base in the

catalytic cavity (*Garron and Cygler, 2010*; *Shaya et al., 2008a*; *Shaya et al., 2006*). The His residue as an *anti*-base can substitute Tyr to absorb the C5 proton in the IdoUA and ensure that β-elimination is carried out successfully, which is detected in chondroitin sulfate ABC lyase (2Q1F) from PL8 family (His[454]) (*Shaya et al., 2008a*) and heparinase II (2FUQ) from PL21 family (His[202]) (*Shaya et al., 2006*), and endows these enzymes with a bifunctional activity against GlcUA- and IdoUA-containing GAGs. Notably, GAGase III, the only identified GAGase with alginate-degrading activity, has a unique His[188] residue also conserved in various alginate lyases from PL5, PL15, PL17, PL38, and PL39 families. Mutation assay showed that the His[188] is a key residue for the alginate-degrading activity of GAGase III. However, when a conversed asparagine residue at the corresponding position of other GAGases was replaced by histidine, the muted GAGases still could not act on alginate, indicating that some other unidentified structural factors are also involved in the alginate degradation of GAGase III. Overall, while the substrate-degrading features of GAGases can be preliminarily elucidated structurally, more direct evidence remains to be obtained by detailed structural and biochemical approaches.

In addition, the identification of GAGases may help clarify an inappropriate naming issue regarding the functional module 'Hepar_II_III superfamily.' This functional module is commonly found in not only GAG lyases but also alginate lyases with an 'N-terminal (α/α)$_n$ toroid + C-terminal β-sandwich' structure. Interestingly, however, a large number of identified alginate lyases with the functional module 'Hepar_II_III superfamily' do not have the ability to degrade Hep/HS. Historically, due to the importance of Hep as an anticoagulant in clinical application, Hep/HS-related studies including heparinases have received much higher attention than alginate-related studies including alginate lyases, which may be the reason why some sequences in alginate lyases that are similar to heparinase are annotated named the 'Hepar_II_III superfamily' module. From an evolutionary perspective, this 'Hepar_II_III superfamily' module might be a conserved functional domain in the divergent evolution process from alginate lyase to GAG lyase. Therefore, it might be more reasonable and accurate for this functional domain to be named as the 'alginate lyase superfamily' module, and the presence of this functional module in GAGases with evolutionary transitional characteristics may well support this view.

In conclusion, the first determination of the structure of GAGases not only facilitates the elucidation of the catalytic mechanism of PL35 family enzymes, especially the substrate selection mechanism of GAGases, but also provides potential evidence for the hypothesis that GAG lyase evolved from alginate lyase.

## Materials and methods
### Materials
HA from *Streptococcus equi*, CS-C from shark cartilage, alginate sodium from brown algae, polyethylene glycol (PEG) 6000, PEG 400, calcium chloride (CaCl$_2$), tris-(hydroxymethyl) aminomethane (Tris), and hydroxyethyl piperazine ethanesulfonic acid (HEPES) were purchased from Sigma–Aldrich. Lysine, phenylalanine, threonine, leucine, isoleucine, valine, and L-selenomethionine were purchased from Solarbio Co., Ltd. (Beijing, China).

### Heterologous expression and purification of GAGases and GAGase II selenomethionine derivant
The cloning and induced expression of PL35 family GAGases were described in previous study (*Wei et al., 2024*). Briefly, the synthesized expression plasmid pET-30a-GAGases were transformed into *E. coli* BL21 (DE3) cells and culture in LB broth at 37 °C. Recombinant enzymes with a 6His-tag were expressed at 16 °C for 16 hr by inducing with isopropyl 1-thio-β-D-galactopyranosid (IPTG) (final concentration of 0.05 mM) at the cell density of OD$_{600}$=0.6–0.8. Furthermore, to obtain the Se-GAGase II, cultured cells harboring the pET-30a-GAGase II were centrifuged at 3700 g at 4 °C for 5 min, washed with M9 culture medium (17.2 g/L Na$_2$HPO$_4$, 3 g/L KH$_2$PO$_4$, 2.5 g/L NaCl and 5 g/L NH$_4$Cl), transferred into M9 culture medium supplemented with 1 mM MgSO$_4$, 0.1 mM CaCl$_2$, 0.4% (w/v) glucose and cultured at 37 °C until the OD$_{600}$ reached 0.6. Then, the culture medium was cooled to 22 °C. Additional essential amino acids, including lysine (100 mg/L), phenylalanine (100 mg/L), threonine (100 mg/L), leucine (50 mg/L), isoleucine (50 mg/L), valine (50 mg/L), and L-selenomethionine (L-SeMet) (50 mg/L), were added and incubated at 22 °C for 15 min. The proteins contained L-SeMet were also induced and expressed at 16 °C for 16 hr by supplementing with 5 mM IPTG.

After further induced cultivation, cells were harvested, resuspended, and disrupted by sonication, the recombinant proteins in the supernatant of cell lysates are primarily purified by loading on a nickel affinity column, washing with buffer A containing 10 mM imidazole to remove impurities, and finally eluting using buffer A containing 250 mM imidazole. After nickel affinity chromatography, the elutes of GAGase II and GAGase VII were diluted and loaded on a Q-Sepharose FF column (GE Healthcare) eluted with a gradient concentration of NaCl from 0 to 1 M, and the target proteins were desalted and concentrated through ultrafiltration and further sub-fractionated by gel filtration on a Superdex G-200 column (GE Healthcare) for crystal culture. The concentration of each protein was determined using BCA Protein Assay Kit with bovine serum albumin (BSA) as reference protein (Cwbio, Shanghai).

## Activity assay of GAGase II, III, VII, and their variants toward various GAGs and alginate

To evaluate the activity of GAGase II, III, VII, and their variants, CS-C or alginate (30 μg) was treated with each wild-type enzyme or its variant (6 μg) in the optimal buffer (50 mM Tris-HCl, pH 7.0) for 12 hr at 40 °C. In order to evaluate the activity of GAGase III-H188N and GAGase III-H188A in degrading alginate, alginate (30 μg) was treated with each variant (3–30 μg) at 40 °C for 12 hr. After inactivated by boiling for 10 min, the products were analyzed by gel filtration HPLC using a Superdex Peptide 10/300 GL column eluted with 0.20 M $NH_4HCO_3$ at a flow rate of 0.4 ml/min and monitored at 232 nm using a UV detector, and online analysis was conducted using LCsolution version 1.25.

To evaluate the activity of variants from the site-directed mutagenesis of some potential substrate-binding residues of GAGase II, HA (150 μg) was treated with the wild-type or each mutant (6 μg) in the optimal buffer (50 mM Tris-HCl, pH 7.0) at 40 °C for 12 hr. To evaluate the activity of GAGase III-H188N and GAGase III-H188A, HA, CS-C, or HS (150 μg) was treated with GAGase III and each variant (6 μg) at 40 °C for 1 hr. The absorbance of each resultant was determined at 232 nm using a UV spectrophotometer.

## Crystallization, X-ray diffraction, and data collection of GAGase II, GAGase VII, and Se-Met-GAGase II

The purified GAGase II (20 mg/ml) was crystallized in 0.1 M HEPES (pH 7.0), 12% (w/v) PEG 6000 and 0.2 M $CaCl_2$. The purified Se-Met-GAGase II (5 mg/ml) was crystallized in 0.1 M Tris (pH 7.5), 20% (w/v) PEG 6000 and 0.2 M $CaCl_2$. The purified GAGase VII (10 mg/ml) was crystallized in 0.1 M HEPES (pH 7.0) and 36% (v/v) PEG 400. The crystallization of all the above proteins was carried out at 18 °C using hanging drop vapor diffusion method in 24-well plates. Crystals were harvested using nylon loops and soaked in cryoprotectant consisting of 20% glycerol and 80% mother liquor. X-ray diffraction data were collected at Shanghai Synchrotron Radiation Facility (SSRF) BL18U1 beamline. Diffraction data-sets were processed using XDS (*Kabsch, 2010*). Data collection statistics are summarized in *Table 1*.

## Structure determination and refinement

The structure of GAGase II was solved by selenium single-wavelength anomalous dispersion (Se-SAD) using AutoSol from Phenix suite (*Adams et al., 2011*). The structure of GAGase VII was determined by molecular replacement using Phaser (*McCoy et al., 2007*) with the structure of GAGase II as the search model. Initial model building was carried out using AutoBuild from Phenix suite. Then several rounds of refinement and manual building were performed alternately using Phenix.Refine and Coot (*Emsley and Cowtan, 2004*), respectively. The ion type was confirmed by inductively coupled plasma-mass spectrometry (ICP-MS) analysis. For each sample, 10 ml of GAGase II or GAGase VII (10 mg) was mixed with 10 ml of nitric acid in equal proportions. The mixture was then placed in a metal bath at 120 °C and heated until clarified. Subsequently, the clarified mixture was filtered through a 0.22 μm membrane and analyzed using ICP-MS (NexION, PerkinElmer). The structures of other GAGases were predicted using RoseTTAfold (*Baek et al., 2021*). Structure alignments were performed using ChimeraX with point accepted mutation (PAM)–120 mtrix (*Pettersen et al., 2021*). All of the structure illustrations were generated using ChimeraX (https://www.cgl.ucsf.edu/chimerax/) and Pymol (https://www.pymol.org/2/). A structure-based similarity search for GAGase II and VII were performed using the DALI server (http://ekhidna2.biocenter.helsinki.fi/dali/) (*Holm and Rosenström, 2010*). The identification of tunnels and channels in proteins were performed using the Caver Web v1.2 (https://loschmidt.chemi.muni.cz/caverweb/) (*Stourac et al., 2019*). The classification of identified enzymes

was confirmed based on the database of carbohydrate-active enzymes (CAZy) (http://www.cazy.org/) (*Drula et al., 2022*) and the structures of identified PLs were download from the RCSB Protein Data Bank (PDB) (https://www.rcsb.org/) (*Berman et al., 2000*).

## Site-directed mutagenesis of GAGases

Some potentially important residues located in and around the active center were individually replaced with alanine (Ala) by using the Fast Mutagenesis Kit V2 from Vazyme Biotech Co., Ltd (Nanjing, China) and these mutants were expressed and examined for activity against various substrates (HA, CS-C, HS, and alginate) to confirm their crucial role in enzyme catalysis. The primer pairs of these mutants are shown in the *Supplementary file 1c*. The expression and purification of GAGases and each variant were estimated by SDS-polyacrylamide gel electrophoresis (SDS-PAGE) followed by staining with Coomassie Brilliant Blue R-250. The results of SDS-PAGE analysis are shown in *Figure 4—figure supplement 2*.

## Molecular docking

Tertiary structure-defined GAG or alginate hexasaccharide was used to dock into the active cavity of GAGase II by using the Autodock Vina program (*Eberhardt et al., 2021*; *Trott and Olson, 2010*) and Molecular Operating Environment. Docking by using the Autodock Vina program was performed on a search space of 15.00×18.19×19.53 Å covering the active centre (x, y, z=–2.60,–12.57, 34.13). The binding energies using non-sulfated and 4-$O$-trisulfated hexasaccharides are calculated as –5.874 and –5.163 kcal/mol, respectively.

## Acknowledgements

This work was supported by the National Key R&D Program of China (No. 2024YFC2816005), National Natural Science Foundation of China (Nos. 31971201, 32330001, 31570071, 31800665), Major Scientific and Technology Innovation Project (MSTIP) of Shandong Province (No. 2019JZZY010817), Natural Science Foundation of Shandong Province (No. ZR2023MC017), the SKLMT Frontiers and Challenges Project (SKLMTFCP-2023–06), Shandong Provincial Youth Innovation Science and Technology Support Program for Colleges and Universities (2022KJ003).

## Additional information

### Funding

| Funder | Grant reference number | Author |
| --- | --- | --- |
| National Key Research and Development Program of China | 2024YFC2816005 | Fuchuan Li |
| National Natural Science Foundation of China | 31971201 | Fuchuan Li |
| National Natural Science Foundation of China | 32330001 | Yu-Zhong Zhang |
| National Natural Science Foundation of China | 31570071 | Fuchuan Li |
| National Natural Science Foundation of China | 31800665 | Wenshuang Wang |
| Major Scientific and Technology Innovation Project (MSTIP) of Shandong Province | 2019JZZY010817 | Fuchuan Li |
| Natural Science Foundation of Shandong Province | ZR2023MC017 | Wenshuang Wang |

| Funder | Grant reference number | Author |
| --- | --- | --- |
| The SKLMT Frontiers and Challenges Project | SKLMTFCP-2023-06 | Fuchuan Li |
| Shandong Provincial Youth Innovation Science and Technology Support Program for Colleges and Universities | 2022KJ003 | Wenshuang Wang |
| Natural Science Foundation of Shandong Province | ZR2024MD015 | Fuchuan Li |
| State Key Laboratory of Microbial Technology Open Projects Fund | M2024-15 | Fuchuan Li |

The funders had no role in study design, data collection and interpretation, or the decision to submit the work for publication.

### Author contributions

Lin Wei, Hai-Yan Cao, Validation, Investigation, Visualization, Writing - original draft; Ruyi Zou, Min Du, Validation, Investigation; Qingdong Zhang, Xiangyu Xu, Yingying Xu, Wenshuang Wang, Xiu-Lan Chen, Investigation; Danrong Lu, Resources; Yu-Zhong Zhang, Fuchuan Li, Conceptualization, Methodology, Writing – review and editing

### Author ORCIDs

Lin Wei http://orcid.org/0000-0002-9348-9508
Hai-Yan Cao https://orcid.org/0000-0003-1365-2094
Yu-Zhong Zhang https://orcid.org/0000-0002-2017-1005
Fuchuan Li https://orcid.org/0000-0003-1920-9691

Reviewer #1 (Public review): https://doi.org/10.7554/eLife.102422.3.sa1
Reviewer #3 (Public review): https://doi.org/10.7554/eLife.102422.3.sa2
Author response https://doi.org/10.7554/eLife.102422.3.sa3

# Additional files

### Supplementary files

Supplementary file 1. Supplementary data for crystal structure and catalytic mechanism of PL35 family glycosaminoglycan lyases with an ultrabroad substrate spectrum. (a) Sequence information of the identified glycosaminoglycan lyases (GAGases). (b) Inductively coupled plasma-mass spectrometry (ICP-MS) analysis of GAGase II and GAGase VII. (c) Strains and primers used in this study.

MDAR checklist

### Data availability

All data supporting the findings of this study are available within the paper (and supporting information files). Relevant data generated during this study or analyzed in this published article are available from the Mendeley data (DOI: 10.17632/xry23b3xzv.2; DOI: 10.17632/m3fsdjprmt.2). The atomic coordinates and structure factors of the structures in this study have been deposited in the Protein Data Bank (PDB codes: 8KHV and 8KHW). The sequences of GAGase II (SOD82962.1) and GAGase VII (EDV05210.1) are publicly accessible in GenBank.

The following datasets were generated:

| Author(s) | Year | Dataset title | Dataset URL | Database and Identifier |
|---|---|---|---|---|
| Wei L, Cao HY, Li FC | 2024 | The crystal structure of glycosaminoglycan lyase GAGase II | https://www.rcsb.org/structure/8KHV | RCSB Protein Data Bank, 8KHV |
| Wei L, Cao HY, Li FC | 2024 | The crystal structure of glycosaminoglycan lyase GAGase VII | https://www.rcsb.org/structure/8KHW | RCSB Protein Data Bank, 8KHW |
| Li F, Wei L | 2025 | The original data on the enzymatic activity of GAGases and their mutants | https://doi.org/10.17632/xry23b3xzv.1 | Mendeley Data, 10.17632/xry23b3xzv.1 |
| Li F, Wei L | 2025 | The original data on the molecule docking of GAGases and the results of Dali server | https://doi.org/10.17632/m3fsdjprmt.1 | Mendeley Data, 10.17632/m3fsdjprmt.1 |

The following previously published datasets were used:

| Author(s) | Year | Dataset title | Dataset URL | Database and Identifier |
|---|---|---|---|---|
| Sudarsanam P, Ley R, Guruge J, Turnbaugh PJ, Mahowald M, Liep D, Gordon J | 2012 | Heparinase II/III-like protein [Bacteroides intestinalis DSM 17393] | https://www.ncbi.nlm.nih.gov/protein/EDV05210.1 | NCBI Protein, EDV05210.1 |
| Varghese N, Submissions S | 2017 | Heparinase II/III-like protein [Spirosoma fluviale] | https://www.ncbi.nlm.nih.gov/protein/SOD82962.1 | NCBI Protein, SOD82962.1 |

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
