## [Editor Report · eLife Assessment]

This **useful** manuscript reports on the crystal structures of two glycosaminoglycan (GAG) lyases from the PL35 family, along with in vitro enzyme activity assays and comprehensive structure-guided mutagenesis. The authors have addressed key concerns by incorporating additional docking analyses, validating the role of His188 in alginate degradation, and providing ICP-MS data to examine Mn²⁺ binding. While these improvements enhance the study, the study is **incomplete** due to the lack of enzyme-substrate complex structures and reliance on modeling which still limit mechanistic insight. Nonetheless, the revised manuscript presents a more complete analysis that will be of interest to specialists in carbohydrate-active enzymes.

---

## [Referee Report · Reviewer #1 (Public review)]

Summary:

This study aims to uncover molecular and structural details underlying the broad substrate specificity of glycosaminoglycan lyases belonging to a specific family (PL35). They determined the crystal structures of two such enzymes, conducted in vitro enzyme activity assays, and a thorough structure-guided mutagenesis campaign to interrogate the role of specific residues. They made progress towards achieving their aims and I appreciate the attempt of the authors to address my initial comments on the paper.

Impact on the field:

I expect this work will have limited impact on the field, although it does stand on its own as a solid piece of structure-function analysis.

Strengths:

The major strengths of the study were the combination of structure and enzyme activity assays, comprehensive structural analysis, as well as a thorough structure-guided mutagenesis campaign.

Weaknesses:

(Before revision) -the authors claim to have done a ICP-MS experiment to show Mn2+ binds to their enzyme, but did not present the data. The authors could have used the anomalous scattering properties of Mn2+ at the synchrotron to determine the presence and location of this cation (i.e. fluorescence spectra, and/or anomalous data collection at the Mn2+ absorption peak).

*comment after revision: I appreciate that the authors included this data now, and it looks fine.

(Before revision) -the authors have an over-reliance on molecular docking for understanding the position of substrates bound to the enzyme. The docking analysis performed was cursory at best; Autodock Vina is a fine program but more rigorous software could have been chosen, as well we molecular dynamics simulations. As well the authors do not use any substrate/product-bound structures from the broader PL enzyme family to guide the placement of the substrates in the GAGases, and interpret the molecular docking models.

*comment after revision: the authors used another docking program, which is fine, but did not do any MD analysis or comment on why not. Also maybe it is just me but I still do not see a figure explicitly showing an overlay/superposition of the docking results with crystal structures of similar enzymes with similar ligands. The authors do have a statement in this regard but I believe a figure (e.g. an additional panel on S2) would be very helpful to the reader.

(Before revision)-the conclusion that the structures of GAGase II and VII are most similar to the structures of alginate lyases (Table 2 data), and the authors' reliance on DALI, are both questioned. DALI uses a global alignment algorithm, which when used for multi-domain enzymes such as these tends to result in sub-optimal alignment of active site residues, particularly if the active site is formed between the two domains as is the case here. The authors should evaluate local alignment methods focused on optimization of the superposition of a single domain; these methods may result in a more appropriate alignment of the active site residues, and different alignment statistics. This may influence the overall conclusion of the evolutionary history of these PL35 enzymes.

*comment after revision: I'm not sure the authors understood my suggestion as the reply reiterates the original conclusions. I suggest local structural alignment of *only* the toroid and antiparallel β-sheet domains, not global alignment of both domains, as this would improve the accuracy of the structural similarity conclusions.

(Before revision)-the data on the GAGase III residue His188 is not well interpreted; substitution of this residue clearly impacts HA and HS hydrolysis as well. The data on the impact on alginate hydrolysis is weak, which could be due to the fact that the WT enzyme has poor activity against alginate to start with.

*comment after revision: I appreciate that the authors used higher amounts of H188A variants and still do not see activity on alginate, which strengthens the conclusions regarding this substrate. However this variant also has decreased activity against HS (Figure 5C) and thus H188 appears to be important for more substrates than just alginate. The discussion section should be updated accordingly.

(Before revision)-the authors did not use the words "homology", "homologous", or "homolog" correctly (these terms mean the subjects have a known evolutionary relationship, which may or may not be known in the contexts the authors used these targets); the words "similarity" and "similar" are recommended to be used instead.

*comment after revision: I thank the authors for addressing this.

(Before revision)-the authors discuss a "shorter" cavity in GAGases, which does not make sense, and is not supported by any figure or analysis. I recommend a figure with a surface representation of the various enzymes of interest, with dimensions of the cavity labeled (as a supplemental figure). The authors also do not specifically define what subsites are in the context of this family of enzymes, nor do they specifically label or indicate the location of the subsites on the figures of the GAGase II and IV enzyme structures.

*comment after revision: I thank the authors for improving their figures and text description on this point.

---

## [Referee Report · Reviewer #3 (Public review)]

Summary:

The authors characterized previous substrate specificity of several polysaccharide lyases from family PL35 (CAzy) and discovered their unusually broad substrate specificity, being able to degrade three types of GAGs belonging to HA, CS, and HS classes.

In this study they determined the 3D structures of two lyases from this family and identified several residues essential for substrate degradation. Comparison with lyases from other PL families but having the same fold allowed them to propose an Asn, Tyr and His as essential for catalysis. One of the characterized lyases can also degrade alginate and they established a specific His residue as necessary for activity toward this substrate but not sufficient by itself.

Attempts to obtain crystals with substrate or products were unsuccessful, therefore the authors resorted to modeling substrate into the determined structures. The obtained models led them to propose a catalytic mechanism, that generally reflects previously proposed mechanism for lyases with this fold.

Unfortunately, they have no definitive explanation for a broad specificity for the PL35 lyases but suggest that it is related to a shorter substrate binding cleft with a large open space on the nonreducing end of the substrate.

Strengths:

The determination of 3D structure of two PL35 lyases allows comparing them to other lyases with similar fold. The structures show a shorter substrate binding cleft that might be the reason for broader substrate specificity. Essential roles of several residues in catalysis and/or substrate binding were established by mutagenesis.

Weaknesses:

The main weakness is the lack of the structures of an enzyme-substrate/product complex. While the determined structures confirm the predicted two domain fold with a helical toroid domain and a double beta-sheet domain, the explanation for the broad specificity is lacking, except for suggestion that it has to do with a shorter substrate binding cleft. The enzymatic mechanism is hypothesized based on models rather than supported by experimentally determined structure of the complex.

---

## [Author Response]

The following is the authors’ response to the original reviews.

**Public Reviews:**

**Reviewer #1 (Public review):**
Summary:This study aims to uncover molecular and structural details underlying the broad substrate specificity of glycosaminoglycan lyases belonging to a specific family (PL35). They determined the crystal structures of two such enzymes, conducted in vitro enzyme activity assays, and a thorough structure-guided mutagenesis campaign to interrogate the role of specific residues. They made progress towards achieving their aims but I see significant holes in data that need to be determined and in the authors' analyses.Impact on the field:I expect this work will have a limited impact on the field, although, with additional experimental work and better analysis, this paper will be able to stand on its own as a solid piece of structure-function analysis.Strengths:The major strengths of the study were the combination of structure and enzyme activity assays, comprehensive structural analysis, as well as a thorough structure-guided mutagenesis campaign.Weaknesses:There were several weaknesses, particularly:(1) The authors claim to have done an ICP-MS experiment to show Mn2+ binds to their enzyme but did not present the data. The authors could have used the anomalous scattering properties of Mn2+ at the synchrotron to determine the presence and location of this cation (i.e. fluorescence spectra, and/or anomalous data collection at the Mn2+ absorption peak).

Thank you for your kind comment and suggestion. Many studies utilized ICP-MS for the detection of metal ions within proteins (doi: 10.1016/j.jbc.2023.103047; doi: 10.1074/jbc.RA119.011790), so we utilized this method to determine the type of atoms within GAGases. In the revised manuscript, the data of ICP-MS experiment has been presented in “Supplemental Table S1”

(2) The authors have an over-reliance on molecular docking for understanding the position of substrates bound to the enzyme. The docking analysis performed was cursory at best; Autodock Vina is a fine program but more rigorous software could have been chosen, as well we molecular dynamics simulations. As well the authors do not use any substrate/product-bound structures from the broader PL enzyme family to guide the placement of the substrates in the GAGases, and interpret the molecular docking models.

Thank you for your kind comments. The interaction between the enzyme and ligand should be confirmed by resolving the structure of enzyme-ligand complex. Unfortunately, we tried to prepare the co-crystals of GAGases with various oligosaccharide substrates but ultimately failed. Thus, we tried to use docking to explain the catalytic mechanism of polysaccharide lyases using Autodock Vina although this method may be questionable. In the revised manuscript, we predicted the substrate binding site of GAGase II using Caver Web 1.2 and performed molecular docking near the substrate binding site simultaneously using Molecular Operating Environment (MOE) to verify the accuracy of the docking results (Figure 6, Supplemental Figure S4). In addition, a series of enzyme-substrate complex structures of identified PL family enzymes with structural similarities to the GAGases are showed in Supplemental Figure S2, and the positions of the catalytic cavities and the substrate binding modes are similar to those of the molecular docking results, which may also corroborate the referability of our molecular docking results in another aspect.

(3) The conclusion that the structures of GAGase II and VII are most similar to the structures of alginate lyases (Table 2 data), and the authors' reliance on DALI, are both questioned. DALI uses a global alignment algorithm, which when used for multi-domain enzymes such as these tends to result in sub-optimal alignment of active site residues, particularly if the active site is formed between the two domains as is the case here. The authors should evaluate local alignment methods focused on the optimization of the superposition of a single domain; these methods may result in a more appropriate alignment of the active site residues and different alignment statistics. This may influence the overall conclusion of the evolutionary history of these PL35 enzymes.

Thank you for your kind question. As your suggestion, multiple structural alignment assays were carried out for the (α/α)_n_ toroid and the antiparallel β-sheet domain, respectively, based on the structures of GAGs/alginate lyases from PL5, PL8, PL12, PL15, PL17, PL21, PL23, PL36, PL38 and PL39 families. The results showed that the overall structure of GAGases is more similarity to that of PL15, PL17 and PL39 family alginate lyases, which have an (α/α)_6_ toroid and an antiparallel β-sheet domain (Table 3). In terms of the toroid and antiparallel β-sheet domains, most of them have an (α/α)_6_ toroid and an antiparallel β-sheet as shown in Table 3. We also noticed that GAGases possess such a (α/α)_6_ toroid structure rather than a (α/α)_7_ toroid structure, and revised the relevant statement in the manuscript.

(4) The data on the GAGase III residue His188 is not well interpreted; substitution of this residue clearly impacts HA and HS hydrolysis as well. The data on the impact on alginate hydrolysis is weak, which could be due to the fact that the WT enzyme has poor activity against alginate to start with.

Thank you very much for your helpful comments and questions. To verify your suggestion that the weak impact of alginate hydrolysis could be due to poor activity of wild type GAGase III, we degraded alginate using different enzyme concentrations (3 to 30 μg) and analyzed the degradation products. The results showed that the alginate-degrading activity of GAGase III-H188A and GAGase III-H188N was abolished, even at a quite high ratio of the mutated enzyme to substrate such as 30 μg enzyme to 30 μg substrate (Supplemental Figure S3A), while their GAG-degrading activity was only partially affected, indicating that this residue plays a more important role for the digestion of alginate than other substrates. Unfortunately, we were unable to confer the ability to GAGase III through the mutation of N191H in GAGase II. Therefore, we suggest that His^188^ play a key role in the specificity of alginate degradation by GAGase III, but that other determinants also contribute to this process. We will try more methods to obtain the structure of enzyme-substrate co-crystals and explain its substrate-selective mechanism in future studies.

(5) The authors did not use the words "homology", "homologous", or "homolog" correctly (these terms mean the subjects have a known evolutionary relationship, which may or may not be known in the contexts the authors used these targets); the words "similarity" and "similar" are recommended to be used instead.

Thank you for your helpful suggestions. We have revised the relevant part of the description in the manuscript.

(6) The authors discuss a "shorter" cavity in GAGases, which does not make sense and is not supported by any figure or analysis. I recommend a figure with a surface representation of the various enzymes of interest, with dimensions of the cavity labeled (as a supplemental figure). The authors also do not specifically define what subsites are in the context of this family of enzymes, nor do they specifically label or indicate the location of the subsites on the figures of the GAGase II and IV enzyme structures.

Thank you for your helpful suggestions. Figures (Supplemental Figure S2) with surface representations of the GAGase II and some structurally similar GAGs/alginate lyases with the dimensions of the cavity labeled, were added to the supplementary data as you suggested. Considering the correlation between enzyme specificity and substrate binding sites, we speculated that a shorter substrate binding cavity might allow the enzyme to accommodate a wider variety of substrates, resulting in a smaller restriction of the catalytic cavity to substrate binding, although this speculation needs to be verified by the resolution of the crystal structure of the enzyme-substrate complexes.

**Reviewer #2 (Public review):**
Summary:Wei et al. present the X-ray crystallographic structures of two PL35 family glycosaminoglycan (GAG) lyases that display a broad substrate specificity. The structural data show that there is a high degree of structural homology between these enzymes and GAGases that have previously been structurally characterized. Central to this are the N-terminal (α/α)7 toroid domain and the C-terminal two-layered β-sheet domain. Structural alignment of these novel PL35 lyases with previously deposited structures shows a highly conserved triplet of residues at the heart of the active sites. Docking studies identified potentially important residues for substrate binding and turnover, and subsequent site-directed mutagenesis paired with enzymatic assays confirmed the importance of many of these residues. A third PL35 GAGase that is able to turn over alginate was not crystallized, but a predicted model showed a conserved active site Asn was mutated to a His, which could potentially explain its ability to act on alginate. Mutation of the His into either Ala or Asn abrogated its activity on alginate, providing supporting evidence for the importance of the His. Finally, a catalytic mechanism is proposed for the activity of the PL35 lyases. Overall, the authors used an appropriate set of methods to investigate their claims, and the data largely support their conclusions. These results will likely provide a platform for further studies into the broad substrate specificity of PL35 lyases, as well as for studies into the evolutionary origins of these unique enzymesStrengths:The crystallographic data are of very high quality, and the use of modern structural prediction tools to allow for comparison of GAGase III to GAGase II/GAGase VII was nice to see. The authors were comprehensive in their comparison of the PL35 lyases to those in other families. The use of molecular docking to identify key residues and the use of site-directed mutagenesis to investigate substrate specificity was good, especially going the extra distance to mutate the conserved Asn to His in GAGase II and GAGase VII.Weaknesses:The structural models simply are not complete. A cursory look at the electron density and the models show that there are many positive density peaks that have not had anything modelled into them. The electron density also does not support the placement of a Mn2+ in the model. The authors indicate that ICP-MS was done to identify the metal, but no ICP-MS data is presented in the main text or supplementary. I believe the authors put too much emphasis on the possibility of GAGase III representing an evolutionary intermediate between GAG lyases and alginate lyases based on a single Asn to His mutation in the active site, and I don't believe that enough time was spent discussing how this "more open and shorter" catalytic cavity would necessarily mean that the enzyme could accommodate a broader set of substrates. Finally, the proposed mechanism does not bring the enzyme back to its starting state.
**Recommendations for the authors:**

**Reviewer #1 (Recommendations for the authors):**
Minor points:(1) The number of significant digits used in Table 1 and Figure 3 legend are not justified. The authors should use a maximum of 2 significant digits.

Thank you for your kind suggestion. We have verified the relevant data and retained two significant digits.

(2) The authors should use the words "mutant" or "mutation" only when discussing DNA, but when discussing protein, the words "variant" and "substitution" should be used instead as these are more appropriate.

Thank you for your helpful suggestions. We have revised the relevant description in the manuscript as you suggested.

(3) Lines 102-110 are a long, run-on sentence that should be split into shorter sentences. Similarly, lines 367-378 should be split into shorter sentences.

Thank you for your suggestions. In the revised manuscript, the long sentences in lines 102-110 and 367-378 have been rewritten into shorter ones.

(4) Lines 174-175: His, Tyr, Glu, and Trp are not positively charged residues and this wording should be changed.

Thank you for your suggestions. We have revised the relevant description in the manuscript as you suggested.

(5) Lines 423-426 require a reference.

Thank you for your suggestion. We have provided the reference at the right position and revised the relevant description in the manuscript as you suggested.

(6) Grammar/language:-line 90 - change "should emerge" to "likely emerged"-line 145 - delete "Finally"-line 264 - delete "their"-line 265 - delete "active sites"-line 265-266 - change to "To confirm this hypothesis, site-directed mutagenesis followed by enzyme activity assay was performed"-line 311 - change "residue in the catalytic cavity of GAGase III, which.." to "residue in its catalytic cavity, which..."-line 318 - change "affect" to "affected"-line 323 - change to "degrading activity of GAGase II remains to be determined outside of the His188 residue"-line 345 - delete "assays"-line 359 - change to "evidence"-line 397 - change "folds" to "3D fold"-line 420 - change to "share similar catalytic sites"-lines 411, 433 - change "conversed" to "conserved"-line 441 - change to "Mutational analysis showed that the His188.."-line 450 - delete "which"

Thank you for your suggestions. Grammatical errors in the revised manuscript have been corrected in the revised manuscript.

**Reviewer #2 (Recommendations for the authors):**
Major ConcernsThe electron density in your model clearly does not support the placement of a Mn ion. In the GAGase II structure, the placement of the Mn and the placement of waters around it still results in two density peaks of > 12 rmsd. The manuscript suggests that ICP-MS was done but the results of this are not shown anywhere. Please include your ICP-MS data. I see the structures have already been deposited, and if they have been deposited unchanged, please see if you can modify them to actually finish building the models. I don't find your data in Figure 2B particularly convincing that Mn is necessarily important for activity.

Thank you for your kind comments. As we known, ICP-MS is a common method used for the detection of metal ions within proteins (doi: 10.1016/j.jbc.2023.103047; doi: 10.1074/jbc.RA119.011790), and thus we utilized it to determine the type of atoms within GAGases in this study. In the revised manuscript, the data of ICP-MS experiment has been presented in “Supplemental Table S1”, and the data clearly showed that the content of Mn^2+^ rather than others in test sample is much higher than that in the negative control, suggesting the involvement of Mn^2+^ in the protein. We agree that the addition of Mn^2+^ does not show very strong promotion to the activity of GAGase II just like other tested metal ions, but the addition of EDTA significantly inhibited the enzyme activity (Figure 2), indicating that metal ion such as Mn^2+^ is necessary for the function of GAGases. Regarding the role of metal ion, whether it participates in the catalytic reaction or only stabilize the structure of enzyme remains to be further explored in our further study.

Minor Concerns(1) Please include CC1/2 in your Table 1.

Thank you for your kind suggestions. CC1/2 parameters have been added in the revised manuscript (Table 1).

(2) If possible please include SDS-PAGE gel images of your purified proteins. Particularly for the point mutations. Ideally, you would have done SEC on your mutants to show that the reduction in activity is not due to aggregation/misfolding, but at the very least I would to see that you have similar levels of purity.

Thank you for your kind suggestions. As your suggestion, we have added SDS-PAGE gel images of purified GAGase II, GAGase III, GAGase VII, and their mutant enzymes to the supplementary data. As shown in Figure S5, site-directed mutagenesis did not affect the soluble expression levels of GAGase II, GAGase III or GAGase VII, indicating that the reduction in activity is not due to aggregation or misfolding. Due to the large number of variants, we used crude enzyme for the activity assay of substrate binding sites, while for some catalytic key residues, we purified the corresponding mutant enzymes and then verified their activities by HPLC.

(3) When referring to your structural predictions, it is not appropriate to say that you used Robetta. Your reference is correct though - you should say that the structures were predicted using RoseTTAfold.

Thank you for your helpful suggestions. We have revised the relevant description in the manuscript.

(4) If possible expand on how the shorter/more open active site cavity would result in broader substrate specificity.

Thank you for your kind comment. In the revised manuscript, figures (Supplemental Figure S2) with surface representations of the GAGase II and some representatively structurally similar GAGs/alginate lyases, with the dimensions of the cavity labeled, were added to the supplementary data. Considering the correlation between enzyme specificity and substrate binding sites, we speculated that a shorter substrate binding cavity might allow the enzyme to accommodate a wider variety of substrates, resulting in a smaller restriction of the catalytic cavity to substrate binding. However, unfortunately, we did not succeed in obtaining co-crystals of GAGases with any of the substrates. We will try to explain the mechanism of substrate selectivity in future studies by culturing and resolving crystals of its enzyme substrate complex or otherwise.

(5) I would put less emphasis on His188 in GAGase III being a strong indicator that this protein represents an evolutionary intermediate between alginate lyases and GAGases.

Thank you for your comment. The His^188^ residue, which is unique compared to other GAGases, is essential for the alginate-degrading activity of GAGase III. Regarding why GAGases are thought to represent a possible evolutionary intermediate between alginate lyases and GAG lyases, phylogenetic analysis demonstrated that GAGases show considerable homology with some identified GAG lyases and alginate lyases (DOI: 10.1016/j.jbc.2024.107466). The similarity in primary structure between some GAG lyases, alginate lyases, and GAGases suggests structural similarities, which are further supported by this study. As structure determines function, structural similarity is often used as a key criterion when studying the evolution of proteins, the GAGase III, which shows significant GAGs and alginate-degrading activity, support for this speculation. Of course, in this study, our analysis of the evolutionary relationship between GAGases and identified GAG lyases and alginate lyases, based on structural comparison, is an attempt using existing methods. The conclusions we have drawn remain a hypothesis that still requires further evidence to support and validate.